# NeuroQuery, comprehensive meta-analysis of human brain mapping

Jérôme Dockès[1]*, Russell A Poldrack[2], Romain Primet[3], Hande Gözükan[3], Tal Yarkoni[4], Fabian Suchanek[5], Bertrand Thirion[1], Gaël Varoquaux[1,6]*

[1]Inria, CEA, Université Paris-Saclay, Essonne, France; [2]Stanford University, Stanford, United States; [3]Inria, Paris, France; [4]University of Texas at Austin, Austin, United States; [5]Télécom Paris University, Palaiseau, France; [6]Montréal Neurological Institute, McGill University, Montreal, Canada

**Abstract** Reaching a global view of brain organization requires assembling evidence on widely different mental processes and mechanisms. The variety of human neuroscience concepts and terminology poses a fundamental challenge to relating brain imaging results across the scientific literature. Existing meta-analysis methods perform statistical tests on sets of publications associated with a particular concept. Thus, large-scale meta-analyses only tackle single terms that occur frequently. We propose a new paradigm, focusing on prediction rather than inference. Our multivariate model *predicts* the spatial distribution of neurological observations, given text describing an experiment, cognitive process, or disease. This approach handles text of arbitrary length and terms that are too rare for standard meta-analysis. We capture the relationships and neural correlates of 7547 neuroscience terms across 13 459 neuroimaging publications. The resulting meta-analytic tool, neuroquery.org, can ground hypothesis generation and data-analysis priors on a comprehensive view of published findings on the brain.

## Introduction

### Pushing the envelope of meta-analyses

Each year, thousands of brain-imaging studies explore the links between brain and behavior: more than 6000 publications a year contain the term 'neuroimaging' on PubMed. Finding consistent trends in the knowledge acquired across these studies is crucial, as individual studies by themselves seldom have enough statistical power to establish fully trustworthy results (*Button et al., 2013*; *Poldrack et al., 2017*). But compiling an answer to a specific question from this impressive number of results is a daunting task. There are too many studies to manually collect and aggregate their findings. In addition, such a task is fundamentally difficult due to the many different aspects of behavior, as well as the diversity of the protocols used to probe them.

Meta-analyses can give objective views of the field, to ground a review article or a discussion of new results. Coordinate-Based Meta-Analysis (CBMA) methods (*Laird et al., 2005*; *Wager et al., 2007*; *Eickhoff et al., 2009*) assess the consistency of results across studies, comparing the observed spatial density of reported brain stereotactic coordinates to the null hypothesis of a uniform distribution. Automating CBMA methods across the literature, as in NeuroSynth (*Yarkoni et al., 2011*), enables large-scale analyses of brain-imaging studies, giving excellent statistical power. Existing meta-analysis methods focus on identifying effects reported consistently across the literature, to distinguish true discoveries from noise and artifacts. However, they can only address neuroscience concepts that are easy to define. Choosing which studies to include in a meta-analysis can be challenging. In principle, studies can be manually annotated as carefully as one likes. However, manual meta-analyses are not scalable, and the corresponding degrees of freedom are

*For correspondence:
jerome@dockes.org (JD);
gael.varoquaux@inria.fr (GV)

difficult to control statistically. In what follows, we focus on automated meta-analysis. To automate the selection of studies, the common solution is to rely on terms present in publications. But closely related terms can lead to markedly different meta-analyses (*Figure 1*). The lack of a universally established vocabulary or ontology to describe mental processes and disorders is a strong impediment to meta-analysis (*Poldrack and Yarkoni, 2016*). Indeed, only 30% of the terms contained in a neuroscience ontology or meta-analysis tool are common to another (see *Table 1*). In addition, studies are diverse in many ways: they investigate different mental processes, using different terms to describe them, and different experimental paradigms to probe them (*Newell, 1973*). Yet, current meta-analysis approaches model all studies as asking the same question. They cannot model nuances across studies because they rely on in-sample statistical inference and are not designed to interpolate between studies that address related but different questions, or make predictions for unseen combinations of mental processes. A consequence is that, as we will show, their results are harder to control outside of well-defined and frequently-studied psychological concepts.

Currently, an automated meta-analysis cannot cover all studies that report a particular functional contrast (contrasting mental conditions to isolate a mental process, *Poldrack et al., 2011*). Indeed, we lack the tools to parse the text in articles and reliably identify those that relate to equivalent or very similar contrasts. As an example, consider a study of the neural support of translating orthography to phonology, probed with visual stimuli by *Pinho et al. (2018)*. The results of this study build upon an experimental contrast labeled by the authors as 'Read pseudo-words vs. consonant strings', shown in *Figure 2*. Given this description, what prior hypotheses arise from the literature for this contrast? Conversely, given the statistical map resulting from the experiment, how can one compare it with previous reports on similar tasks? For these questions, meta-analysis seems the tool of choice. Yet, the current meta-analytic paradigm requires the practitioner to select a set of studies that are included in the meta-analysis. In this case, which studies from the literature should be included? Even with a corpus of 14 000 full-text articles, selection based on simple pattern matching –as with NeuroSynth– falls short. Indeed, only 29 studies contain all 5 words from the contrast description, which leads to a noisy and under-powered meta-analytic map (*Figure 2*). To avoid relying on the contrast name, which can be seen as too short and terse, one could do a meta-analysis based on the page-long task description (that can be found at https://project.inria.fr/IBC/data/ and is reproduced in the supplementary data). However, that would require combining even more terms, which precludes selecting studies that contain all of them. A more manual selection may help to identify relevant studies, but it is far more difficult and time-consuming. Moreover, some concepts of interest may not have been investigated by themselves, or only in very few studies: rare diseases, or tasks involving a combination of mental processes that have not been studied together. For instance, there is evidence of agnosia in Huntington's disease (*Sitek et al., 2014*), but it has not been studied with brain imaging. To compile a brain map from the literature for such queries, it is necessary to interpolate between studies only partly related to the query. Standard meta-analytic methods lack an automatic way to measure the relevance of studies to a question, and to interpolate between

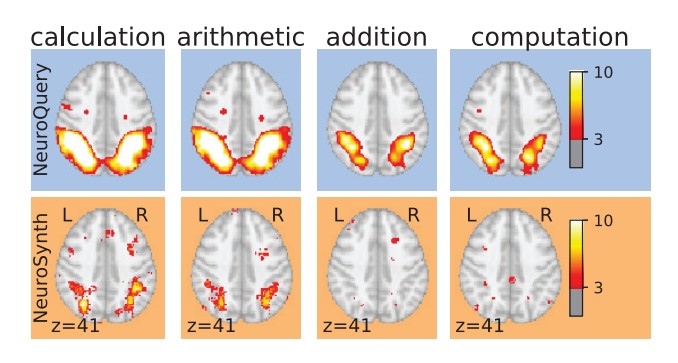

**Figure 1.** Taming query variability Maps obtained for a few words related to mental arithmetic. By correctly capturing the fact that these words are related, NeuroQuery can use its map for easier words like 'calculation' and 'arithmetic' to encode terms like 'computation' and 'addition' that are difficult for meta-analysis.

**Table 1.** Diversity of vocabularies: there is no established lexicon of neuroscience, even in hand-curated reference vocabularies, as visible across CognitiveAtlas (*Poldrack and Yarkoni, 2016*), MeSH (*Lipscomb, 2000*), NeuroNames (*Bowden and Martin, 1995*), NIF (*Gardner et al., 2008*), and NeuroSynth (*Yarkoni et al., 2011*).

Our dataset, NeuroQuery, contains all the terms from the other vocabularies that occur in more than 5 out of 10 000 articles. 'MeSH' corresponds to the branches of PubMed's MEdical Subject Headings related to neurology, psychology, or neuroanatomy (see Section 'The choice of vocabulary'). Many MeSH terms are hardly or never used in practice – For example variants of multi-term expressions with permuted word order such as 'Dementia, Frontotemporal', and are therefore not included in NeuroQuery's vocabulary. Numbers above 25% are shown in bold.

| % of ↓ contained in → | Cognitive Atlas (895) | MeSH (21287) | NeuroNames (7146) | NIF (6912) | NeuroSynth (1310) | NeuroQuery(7547) |
|---|---|---|---|---|---|---|
| Cognitive Atlas | **100%** | 14% | 0% | 3% | 14% | **68%** |
| MeSH | 1% | **100%** | 3% | 4% | 1% | 9% |
| NeuroNames | 0% | 9% | **100%** | **29%** | 1% | 10% |
| NIF | 0% | 12% | **30%** | **100%** | 1% | 10% |
| NeuroSynth | 9% | 14% | 5% | 5% | **100%** | **98%** |
| NeuroQuery | 8% | **25%** | 9% | 9% | 17% | **100%** |

them. This prevents them from answering new questions, or questions that cannot be formulated simply.

Many of the constraints of standard meta-analysis arise from the necessity to define an *in-sample* test on a given set of studies. Here, we propose a new kind of meta-analysis, that focuses on *out-of-sample* prediction rather than hypothesis testing. The focus shifts from establishing consensus for a particular subject of study to building multivariate mappings from mental diseases and psychological concepts to anatomical structures in the brain. This approach is complementary to classic meta-analysis methods such as Activation Likelihood Estimate (ALE) (*Laird et al., 2005*), Multilevel Kernel Density Analysis (MKDA) (*Wager et al., 2007*) or NeuroSynth (*Yarkoni et al., 2011*): these perform statistical tests to evaluate trustworthiness of results from past studies, while our framework predicts, based on the description of an experiment or subject of study, which brain regions are most likely to be observed in a study. We introduce a new meta-analysis tool, NeuroQuery, that predicts the

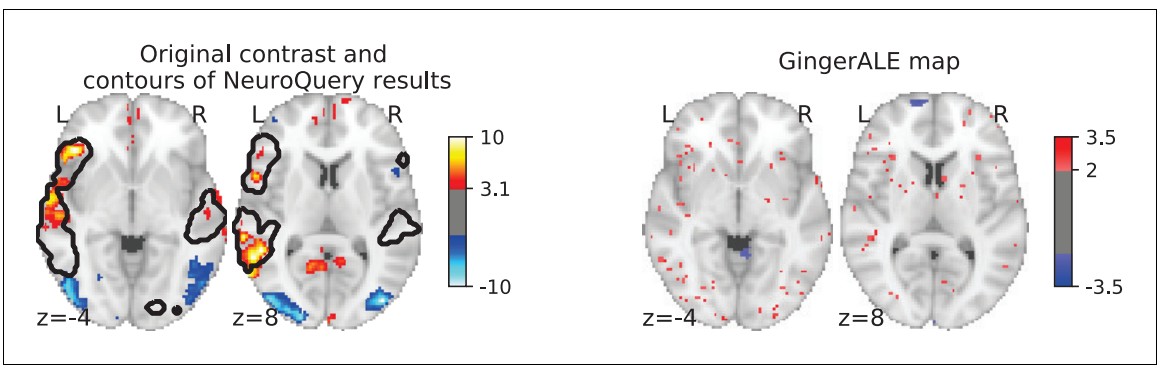

**Figure 2.** Illustration: studying the contrast 'Read pseudo words *vs.* consonant strings'. Left: Group-level map from the IBC dataset for the contrast 'Read pseudo-words vs. consonant strings' and contour of NeuroQuery map obtained from this query. The NeuroQuery map was obtained directly from the contrast description in the dataset's documentation, without needing to manually select studies for the meta-analysis nor convert this description to a string pattern usable by existing automatic meta-analysis tools. The map from which the contour is drawn, as well as a NeuroQuery map for the page-long description of the RSVP language task, are shown in Section 'Example Meta-analysis results for the RSVP language task from the IBC dataset', in Section 'Example Meta-analysis results for the RSVP language task from the IBC dataset' c and Section 'Example Meta-analysis results for the RSVP language task from the IBC dataset'd respectively. Right: ALE map for 29 studies that contain all terms from the IBC contrast description. The map was obtained with the GingerALE tool (*Eickhoff et al., 2009*). With only 29 matching studies, ALE lacks statistical power for this contrast description.

neural correlates of neuroscience concepts – related to behavior, diseases, or anatomy. To do so, it considers terms not in isolation, but in a dynamic, contextually-informed way that allows for mutual interactions. A predictive framework enables maps to be generated by generalizing from terms that are well studied ('faces') to those that are less well studied and inaccessible to traditional meta-analyses ('prosopagnosia'). As a result, NeuroQuery produces high-quality brain maps for concepts studied infrequently in the literature and for a larger class of queries than existing tools – including, for example free text descriptions of a hypothetical experiment. These brain maps predict well the spatial distribution of findings and thus form good grounds to generate regions of interest or interpret results for studies of infrequent terms such as prosopagnosia. Yet, unlike with conventional meta-analysis, they do not control a voxel-level null hypothesis, hence are less suited to asserting that a particular area is activated in studies, for example of prosopagnosia.

Our approach, NeuroQuery, assembles results from the literature into a brain map using an arbitrary query with words from our vocabulary of 7547 neuroscience terms. NeuroQuery uses a multivariate model of the statistical link between multiple terms and corresponding brain locations. It is fitted using supervised machine learning on 13459 full-text publications. Thus, it learns to weight and combine terms to predict the brain locations most likely to be reported in a study. It can predict a brain map given any combination of terms related to neuroscience – not only single words, but also detailed descriptions, abstracts, or full papers. With an extensive comparison to published studies, we show in Section 'Quantitative evaluation: NeuroQuery is an accurate model of the literature' that it indeed approximates well results of actual experimental data collection. NeuroQuery also models the semantic relations that underlie the vocabulary of neuroscience. Using techniques from natural language processing, NeuroQuery infers semantic similarities across terms used in the literature. Thus, it makes better use of the available information, and can recover biologically plausible brain maps where other automated methods lack statistical power, for example with terms that are used in few studies, as shown in Section 'NeuroQuery can map rare or difficult concepts'. This semantic model also makes NeuroQuery less sensitive to small variations in terminology (*Figure 1*). Finally, the semantic similarities captured by NeuroQuery can help researchers navigate related neuroscience concepts while exploring their associations with brain activity. NeuroQuery extends the scope of standard meta-analysis, as it extracts from the literature a comprehensive statistical summary of evidence accumulated by neuroimaging research. It can be used to explore the domain knowledge across sub-fields, generate new hypotheses, and construct quantitative priors or regions of interest for future studies, or put in perspective results of an experiment. NeuroQuery is easily usable online, at neuroquery.org, and the data and source code can be freely downloaded. We start by briefly describing the statistical model behind NeuroQuery in Section 'Overview of the NeuroQuery model', then illustrate its usage (Section 'Illustration: using NeuroQuery for post-hoc interpretation') and show that it can map new combinations of concepts in Section 'NeuroQuery can map new combinations of concepts'. In Section 'NeuroQuery can map rare or difficult concepts and Quantitative evaluation: NeuroQuery is an accurate model of the literature', we conduct a thorough qualitative and quantitative assessment of the new possibilities it offers, before a discussion and conclusion.

## Results

### The NeuroQuery tool and what it can do
#### Overview of the NeuroQuery model
NeuroQuery is a statistical model that identifies brain regions related to an arbitrary text query – a single term, a few keywords, or a longer text. It is built on a controlled vocabulary of neuroscience terms and a large corpus containing the full text of neuroimaging publications and the coordinates that they report. The main components of the NeuroQuery model are an estimate of the relatedness of terms in the vocabulary, derived from co-occurrence statistics, and a regression model that links term occurrences to neural activations using supervised machine learning techniques. To generate a brain map, NeuroQuery first uses the estimated semantic associations to map the query onto a set of keywords that can be reliably associated with brain regions. Then, it transforms the resulting representation into a brain map using a linear regression model (*Figure 3*). This model can thus be understood as a reduced rank regression, where the low-dimensional representation is a distribution

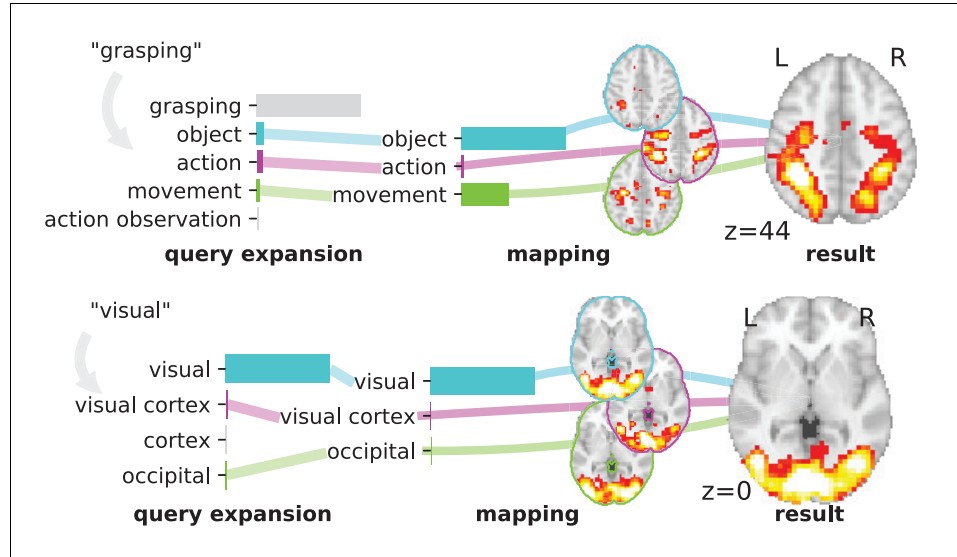

**Figure 3.** Overview of the NeuroQuery model: two examples of how association maps are constructed for the terms 'grasping' and 'visual'. The query is expanded by adding weights to related terms. The resulting vector is projected on the subspace spanned by the smaller vocabulary selected during supervised feature selection. Those well-encoded terms are shown in color. Finally, it is mapped onto the brain space through the regression model. When a word (e.g., 'visual') has a strong association with brain activity and is selected as a regressor, the smoothing has limited effect. Details: the first bar plot shows the semantic similarities of neighboring terms with the query. It represents the smoothed TFIDF vector. Terms that are not used as features for the supervised regression are shown in gray. The second bar plot shows the similarities of selected terms, rescaled by the norms of the corresponding regression coefficient maps. It represents the relative contribution of each term in the final prediction. The coefficient maps associated with individual terms are shown next to the bar plot. These maps are combined linearly to produce the prediction shown on the right.

of weights over keywords selected for their strong link with brain activity. We emphasize the fact that NeuroQuery is a *predictive* model. The maps it outputs are predictions of the likelihood of observation brain location (rescaled by their standard deviation). They do not have the same meaning as ALE, MKDA or NeuroSynth maps as they do not show a voxel-level test statistic. In this section we describe our neuroscience corpus and how we use it to estimate semantic relations, select keywords, and map them onto brain activations.

NeuroQuery relies on a corpus of 13459 full-text neuroimaging publications, described in Section 'Building the NeuroQuery training data'. This corpus is by far the largest of its kind; the NeuroSynth corpus contains a similar number of documents, but uses only the article abstracts, and not the full article texts. We represent the text of a document with the (weighted) occurrence frequencies of each phrase from a fixed vocabulary, that is Term Frequency · Inverse Document Frequency (TFIDF) features (*Salton and Buckley, 1988*). This vocabulary is built from the union of terms from several ontologies (shown in *Table 1*) and labels from 12 anatomical atlases (listed in Table 4 in Section 'The choice of vocabulary'). It comprises 7547 terms or phrases related to neuroscience that occur in at least 0.05% of publications. We automatically extract 418772 peak activations coordinates from publications, and transform them to brain maps with a kernel density estimator. Coordinate extraction is discussed and evaluated in Section 'coordinate extraction'. This preprocessing step thus yields, for each article: its representation in term frequency space (a TFIDF vector), and a brain map representing the estimated density of activations for this study. The corresponding data is also openly available online.

The first step of the NeuroQuery pipeline is a semantic smoothing of the term-frequency representations. Many expressions are challenging for existing automated meta-analysis frameworks, because they are too rare, polysemic, or have a low correlation with brain activity. Rare words are problematic because peak activation coordinates are a very weak signal: from each article we extract little information about the associated brain activity. Therefore existing frameworks rely on the

occurrence of a term in hundreds of studies in order to detect a pattern in peak activations. Term co-occurrences, on the other hand, are more consistent and reliable, and capture semantic relationships (*Turney and Pantel, 2010*). The strength of these relationships encodes semantic proximity, from very strong for synonyms that occur in statistically identical contexts, to weaker for different yet related mental processes that are often studied one opposed to the other. Using them helps meta analysis: it would require hundreds of studies to detect a pattern in locations reported for 'aphasia', for example in lesion studies. But with the text of a few publications we notice that it often appears close to 'language', which is indeed a related mental process. By leveraging this information, Neuro-Query recovers maps for terms that are too rare to be mapped reliably with standard automated meta-analysis. Using Non-negative Matrix Factorization (NMF), we compute a low-rank approximation of word co-occurrences (the covariance of the TFIDF features), and obtain a denoised semantic relatedness matrix (details are provided in Section 'smoothing: regularization at test time'). These word associations guide the encoding of rare or difficult terms into brain maps. They can also be used to explore related neuroscience concepts when using the NeuroQuery tool.

The second step from a text query to a brain map is NeuroQuery's text-to-brain encoding model. When analyzing the literature, we fit a linear regression to reliably map text onto brain activations. The intensity (across the peak density maps) of each voxel in the brain is regressed on the TFIDF descriptors of documents. This model is an additive one across the term occurrences, as opposed to logical operations traditionally used to select studies for meta-analysis. It results in higher predictive power (Section 'Word occurrence frequencies across the corpus').

One challenge is that TFIDF representations are sparse and high-dimensional. We use a reweighted ridge regression and feature selection procedure (described in Section 'reweighted ridge matrix and feature (vocabulary) selection') to prevent uninformative terms such as 'magnetoencephalography' from degrading performance. This procedure automatically selects around 200 keywords that display a strong statistical link with brain activity and adapts the regularization applied to each feature. Indeed, mapping too many terms (covariates) without appropriate regularization would degrade the regression performance due to multicolinearity.

To make a prediction, NeuroQuery combines semantic smoothing and linear regression of brain activations. To encode a new document or query, the text is expanded, or smoothed, by adding weight to related terms using the semantic similarity matrix. The resulting smoothed representation is projected onto the reduced vocabulary of selected keywords, then mapped onto the brain through the linear regression coefficients (*Figure 3*). The rank of this linear model is therefore the size of the restricted vocabulary that was found to be reliably mapped to the brain. Compared with other latent factor models, this 2-layer linear model is easily interpretable, as each dimension (both of the input and the latent space) is associated with a term from our vocabulary. In addition, Neuro-Query uses an estimate of the voxel-level variance of association (see methodological details in Section 'Mathematical details of the NeuroQuery statistical model'), and reports a map of Z statistics. Note that this variance represents an uncertainty around a *prediction* for a TFIDF representation of the concept of interest, which is treated as a fixed quantity. Therefore, the resulting map cannot be thresholded to reject any simple null hypothesis. NeuroQuery maps have a different meaning and different uses than standard meta-analysis maps obtained e.g. with ALE.

## Illustration: using NeuroQuery for post-hoc interpretation

After running a functional Magnetic Resonance Imaging (fMRI) experiment, it is common to compare the computed contrasts to what is known from the existing literature, and even use prior knowledge to assess whether some activations are not specific to the targeted mental process, but due to experimental artifacts such as the stimulus modality. It is also possible to introduce prior knowledge earlier in the study and choose a Region of Interest (ROI) before running the experiment. This is usually done based on the expertise of the researcher, which is hard to formalize and reproduce. With NeuroQuery, it is easy to capture the domain knowledge and perform these comparisons or ROI selections in a principled way.

As an example, consider again the contrast from the RSVP language task (*Pinho et al., 2018*; *Humphries et al., 2006*) in the Individual Brain Charting (IBC) dataset, shown in *Figure 2*. It is described as 'Read pseudo-words vs. consonant strings'. We obtain a brain map from NeuroQuery by simply transforming the contrast description, without any manual intervention, and compare both

maps by overlaying a contour of the NeuroQuery map on the actual IBC group contrast map. We can also obtain a meta-analytic map for the whole RSVP language task by analyzing the free-text task description with NeuroQuery (Section 'Example Meta-analysis results for the RSVP language task from the IBC dataset.').

## NeuroQuery can map new combinations of concepts

To study the predictions of NeuroQuery, we first demonstrate that it can indeed give good brain maps on combinations of terms that have never been studied together. For this, we leave out from our corpus of studies all the publications that simultaneously mention two given terms, we fit a NeuroQuery model on the resulting reduced corpus, and evaluate its predictions on the left out publications, that did actually report these terms together. *Figure 4* shows an example of such an experiment: excluding publications mentioning simultaneously 'distance' and 'color'. The figure compares a simple meta analysis of the combination of these two terms – contrasting the left-out studies with the remaining ones – with the predictions of the model fitted excluding studies that include the term conjunction. Qualitatively, the predicted maps comprise all the brain structures visible in the simultaneous studies of 'distance' and 'color': on the one hand, the intra-parietal sulci, the frontal eye fields, and the anterior cingulate/anterior insula network associated with distance perception, and on the other hand, the additional mid-level visual region around the approximate location of V4 associated with color perception. The extrapolation from two terms for which the model has seen studies, 'distance' and 'color', to their combination, for which the model has no data, is possible thanks to the linear additive model, combining regression maps for 'distance' and 'color'.

To assert that the good generalization to unseen pairs of terms is not limited to the above pair, we apply quantitative experiments of prediction quality (introduced later, in Section 'Quantitative evaluation: NeuroQuery is an accurate model of the literature') to 1 000 randomly-chosen pairs. We find that measures of how well predictions match the literature decrease only slightly for studies with terms already seen together compared to studies with terms never seen jointly (details in Section 'NeuroQuery performance on unseen pairs of terms'). Finally, we gauge the quality of the maps with a quantitative experiment mirroring the qualitative evaluation of *Figure 4*: for each of the 1 000 pairs of terms, we compute the Pearson correlation of the predicted map for the unseen combination of terms with the meta-analytic map obtained on the left-out studies. We find a median correlation of 0.85 which shows that the excellent performance observed on *Figure 4* is not due to a specific choice of terms.

## NeuroQuery can map rare or difficult concepts

We now we compare the NeuroQuery model to existing automated meta-analysis methods, investigate how it handles terms that are challenging for the current state of the art, and quantitatively evaluate its performance. We compare NeuroQuery with NeuroSynth (*Yarkoni et al., 2011*), the best known automated meta-analytic tool, and with Generalized Correspondence Latent Dirichlet Allocation (GCLDA) (*Rubin et al., 2017*). GCLDA is an important baseline because it is the only multivariate meta-analytic model to date. However, it produces maps with a low spatial resolution because it models brain activations as a mixture of Gaussians. Moreover, it takes several days to

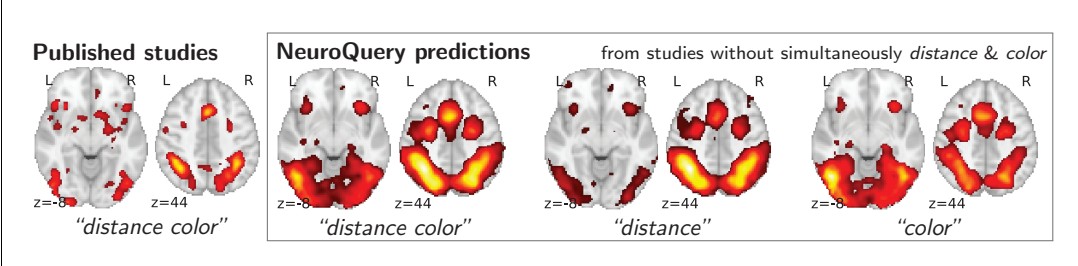

**Figure 4.** Mapping an unseen combination of terms. Left The difference in the spatial distribution of findings reported in studies that contains both 'distance' and 'color' ($n = 687$), and the rest of the studies. – Right Predictions of a NeuroQuery model fitted on the studies that do not contain simultaneously both terms 'distance' and 'color'.

train and a dozen of seconds to produce a map at test time, and is thus unsuitable to build an online and responsive tool like NeuroSynth or NeuroQuery.

By combining term similarities and an additive encoding model, NeuroQuery can accurately map rare or difficult terms for which standard meta-analysis lacks statistical power, as visible on *Figure 5*.

Quantitatively comparing methods on very rare terms is difficult for lack of ground truth. We therefore conduct meta-analyses on subsampled corpora, in which some terms are made artificially rare, and use the maps obtained from the full corpus as a reference. We choose a set of frequent and well-mapped terms, such as 'language', for which NeuroQuery and NeuroSynth (trained on a full corpus) give consistent results. For each of those terms, we construct a series of corpora in which the word becomes more and more rare: from a full corpus, we erase randomly the word from many documents until it occurs at most in $2^{13}$ = 8912 articles, then $2^{12}$ = 4096, and so on. For many terms, NeuroQuery only needs a dozen examples to produce maps that are qualitatively and quantitatively close to the maps it obtains for the full corpus – and to NeuroSynth's full-corpus maps. NeuroSynth typically needs hundreds of examples to obtain similar results, as seen in *Figure 6*. Document frequencies roughly follow a power law (*Piantadosi, 2014*), meaning that most words are very rare – half the terms in our vocabulary occur in less than 76 articles (see Section 'Word occurrence frequencies across the corpus'). Reducing the number of studies required to map well a term (a.k.a. the sample complexity of the meta-analysis model) therefore greatly widens the vocabulary that can be studied by meta-analysis.

Capturing relations between terms is important because the literature does not use a perfectly consistent terminology. The standard solution is to use expert-built ontologies (*Poldrack and Yarkoni, 2016*), but these tend to have low coverage. For example, the controlled vocabularies that we use display relatively small intersections, as can be seen in *Table 1*. In addition, ontologies are typically even more incomplete in listing relations across terms. Rather than ontologies, NeuroQuery relies on distributional semantics and co-occurrence statistics across the literature to estimate relatedness between terms. These continuous semantic links provide robustness to inconsistent terminology: consistent meta-analytic maps for similar terms. For instance, 'calculation', 'computation', 'arithmetic', and 'addition' are all related terms that are associated with similar maps by NeuroQuery. On the contrary, standard automated meta-analysis frameworks map these terms in isolation, and thus suffer from a lack of statistical power and produce empty, or nearly empty, maps for some of these terms (see *Figure 1*).

NeuroQuery improves mapping not only for rare terms that are variants of concepts widely studied, but also for some concepts rarely studied, such as 'color' or 'Huntington' (*Figure 5*). The main reason is the semantic smoothing described in Section 'Overview of the NeuroQuery model'.

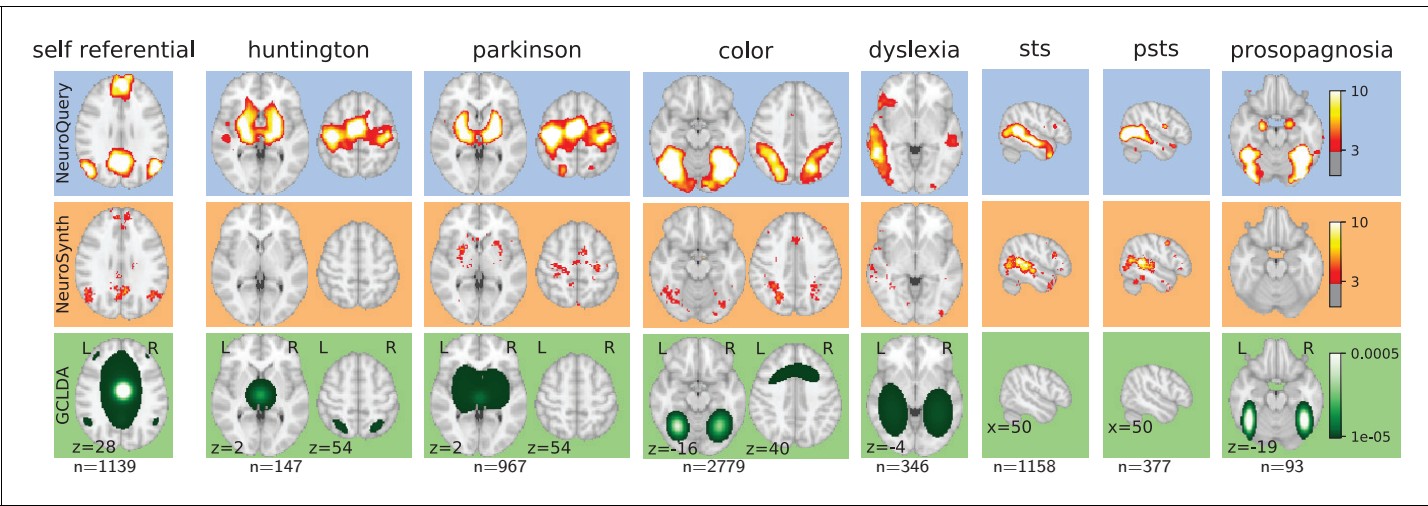

**Figure 5.** Examples of maps obtained for a given term, compared across different large-scale meta-analysis frameworks. 'GCLDA' has low spatial resolution and produces inaccurate maps for many terms. For relatively straightforward terms like 'psts' (posterior superior temporal sulcus), NeuroSynth and NeuroQuery give consistent results. For terms that are more rare or difficult to map like 'dyslexia', only NeuroQuery generates usable brain maps.

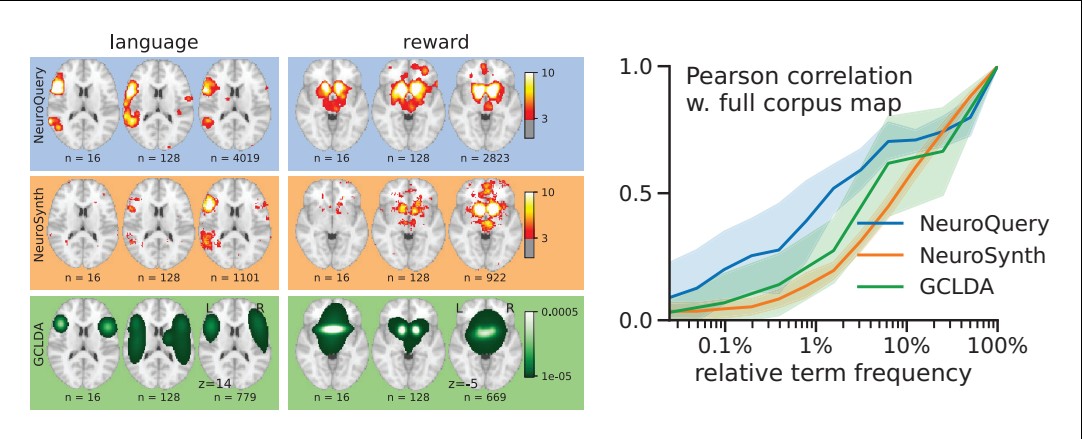

**Figure 6.** Learning good maps from few studies. left: maps obtained from subsampled corpora, in which the encoded word appears in 16 and 128 documents, and from the full corpus. NeuroQuery needs less examples to learn a sensible brain map. NeuroSynth maps correspond to NeuroSynth's Z scores for the 'association test' from neurosynth.org. NeuroSynth's 'posterior probability' maps for these terms for the full corpus are shown in *Figure 19*. Each tool is trained on its own dataset, which is why the full-corpus occurrence counts differ. right: convergence of maps toward their value for the full corpus, as the number of occurrences increases. Averaged over 13 words: 'language', 'auditory', 'emotional', 'hand', 'face', 'default mode', 'putamen', 'hippocampus', 'reward', 'spatial', 'amygdala', 'sentence', 'memory'. On average, NeuroQuery is closer to the full-corpus map. This confirms quantitatively what we observe for the two examples 'language' and 'reward' on the left. Note that here convergence is only measured with respect to the model's own behavior on the full corpus, hence a high value does not indicate necessarily a good face validity of the maps with respect to neuroscience knowledge. The solid line represents the mean across the 13 words and the error bands represent a 95% confidence interval based on 1 000 bootstrap repetitions.

Another reason is that working with the full text of publications associates many more studies to a query: 2779 for 'color', while NeuroSynth matches only 236 abstracts, and 147 for 'huntington', a term not known to NeuroSynth. Full-text matching however requires to give unequal weight to studies, to avoid giving too much weight to studies weakly related to the query. These weights are computed by the supervised-learning ridge regression: in its dual formulation, ridge regression is seen as giving weights to training samples (*Bishop, 2006*, sec 6.1).

## Quantitative evaluation: NeuroQuery is an accurate model of the literature

Unlike standard meta-analysis methods, which compute in-sample summary statistics, NeuroQuery is a predictive model, that can produce brain maps for out-of-sample neuroimaging studies. This enables us to quantitatively assess its generalization performance. Here we check that NeuroQuery captures reliable links from concepts to brain activity – associations that generalize to new, unseen neuroimaging studies. We do this with 16-fold shuffle-split cross-validation. After fitting a NeuroQuery model on 90% of the corpus, for each document in the left-out test set (around 1 300), we encode it, normalize the predicted brain map to coerce it into a probability density, and compute the average log-likelihood of the coordinates reported in the article with respect to this density. The procedure is then repeated 16 times and results are presented in *Figure 7*. We also perform this procedure with NeuroSynth and GCLDA. NeuroSynth does not perform well for this test. Indeed, the NeuroSynth model is designed for single-phrase meta-analysis, and does not have a mechanism to combine words and encode a full document. Moreover, it is a tool for in-sample statistical inference, which is not well suited for out-of sample prediction. GCLDA performs significantly better than chance, but still worse than a simple ridge regression baseline. This can be explained by the unrealistic modelling of brain activations as a mixture of a small number of Gaussians, which results in low spatial resolution, and by the difficulty to perform posterior inference for GCLDA. Another metric, introduced in *Mitchell et al. (2008)* predicting for encoding models, tests the ability of the meta-analytic model to match the text of a left-out study with its brain map. For each article in the test set, we draw randomly another one and check whether the predicted map is closer to the correct map (containing peaks at each reported location) or to the random negative example. More than 72% of the time, NeuroQuery's output has a higher Pearson correlation with the correct map than with the negative example (see *Figure 7* right).

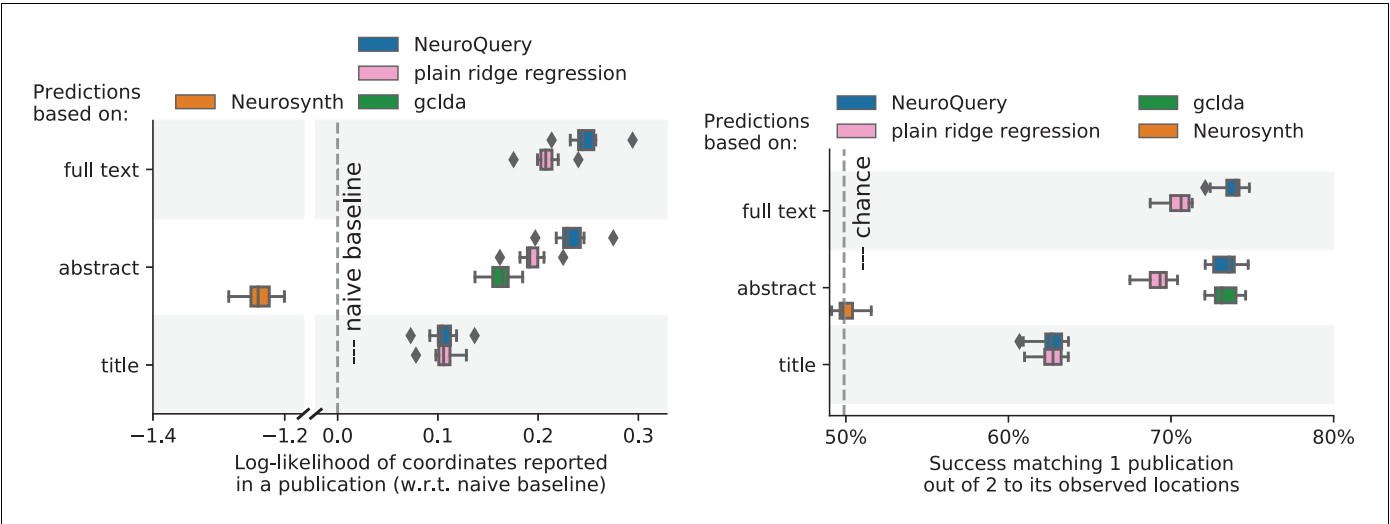

**Figure 7.** Explaining coordinates reported in unseen studies. Left: log-likelihood for coordinates reported in test articles, relative to the log-likelihood of a naive baseline that predicts the average density of the training set. NeuroQuery outperforms GCLDA, NeuroSynth, and a ridge regression baseline. Note that NeuroSynth is not designed to make encoding predictions for full documents, which is why it does not perform well on this task. – Right: how often the predicted map is closer to the true coordinates than to the coordinates for another article in the test set (**Mitchell et al., 2008**). The boxes represent the first, second and third quartiles of scores across 16 cross-validation folds. Whiskers represent the rest of the distribution, except for outliers, defined as points beyond 1.5 times the IQR past the low and high quartiles, and represented with diamond fliers.

## NeuroQuery maps are close to reference meta-analytic maps and atlases

The above experiments quantify how well NeuroQuery captures the information in the literature, by comparing predictions to reported coordinates. However, the scores are difficult to interpret, as peak coordinates reported in the literature are noisy and incomplete with respect to the full activation maps. We also want to quantify the quality of the brain maps generated by NeuroQuery, extending the visual comparisons of *Figure 5*. For this purpose, we compare NeuroQuery predictions to a few reliable references.

First, we use a set of diverse and curated Coordinate-Based Meta-Analysis (IBMA) maps available publicly (**Varoquaux et al., 2018**). This collection contains 19 IBMA brain maps, labelled with cognitive concepts such as 'visual words'. For each of these labels, we obtain a prediction from NeuroQuery and compare it to the corresponding IBMA map. The IBMA maps are thresholded. We evaluate whether thresholding the NeuroQuery predicted maps can recover the above-threshold voxels in the IBMA, quantifying false detections and misses for all thresholds with the Area Under the Receiver Operating Characteristic (ROC) Curve (**Fawcett, 2006**). NeuroQuery predictions match well the IBMA results, with a median Area Under the Curve (AUC) of 0.80. Such results cannot be directly obtained with NeuroSynth, as many labels are missing from NeuroSynth's vocabulary. Manually reformulating the labels to terms from NeuroSynth's vocabulary gives a median AUC of .83 for NeuroSynth, and also raises the AUC to .88 for NeuroQuery (details in Section 'Comparison with the BrainPedia IBMA study' and Figure 13).

We also perform a similar experiment for anatomical terms, relying on the Harvard-Oxford structural atlases (**Desikan et al., 2006**). Both NeuroSynth and NeuroQuery produce maps that are close to the atlases' manually segmented regions, with a median AUC of 0.98 for NeuroQuery and 0.95 for NeuroSynth, for the region labels that are present in NeuroSynth's vocabulary. Details are provided in Section 'Comparison with Harvard-Oxford anatomical atlas' and Figure 14.

For frequent-enough terms, we consider NeuroSynth as a reference. Indeed, while the goal of NeuroSynth is to perform a voxel-level test of independence, and not to predict an activation distribution like NeuroQuery, in most casesNeuroQuery should predict few observations where the test statistic is small. We threshold NeuroSynth maps by controlling the False Discovery Rate (FDR) at 1%

and select the 200 maps with the largest number of activations. We compare NeuroQuery predictions to NeuroSynth activations by computing the AUC. NeuroQuery and NeuroSynth maps for these well-captured terms are very similar, with a median AUC of 0.90. Details are provided in Section 'Comparison with NeuroSynth on terms with strong activations' and Figure 15.

### NeuroQuery is an openly available resource

NeuroQuery can easily be used online: https://neuroquery.org. Users can enter free text in a search box (rather than select a single term from a list as is the case with existing tools) and discover which terms, neuroimaging publications, and brain regions are related to their query. NeuroQuery is also available as an open-source Python package that can be easily installed on all platforms: https://github.com/neuroquery/neuroquery (copy archived at https://github.com/elifesciences-publications/neuroquery). This will enable advanced users to run extensive meta-analysis with Neuroquery, integrate it in other applications, and extend it. The package allows training new NeuroQuery models as well as downloading and using a pre-trained model. Finally, all the resources used to build NeuroQuery are freely available at https://github.com/neuroquery/neuroquery_data (copy archived at https://github.com/elifesciences-publications/neuroquery_data). This repository contains (*i*) the data used to train the model: vocabulary list and document frequencies, word counts (TFIDF features), and peak activation coordinates for our whole corpus of 13 459 publications, (*ii*) the semantic-smoothing matrix, that encodes relations across the terminology. The corpus is significantly richer than NeuroSynth, the largest corpus to date (see Table 3 for a comparison), and manual quality assurance reveals more accurate extraction of brain coordinates (Table 2).

## Discussion

NeuroQuery makes it easy to perform meta-analyses of arbitrary questions on the human neuroscience literature: it uses a full-text description of the question and the studies and it provides an online query interface with a rich database of studies. For this, it departs from existing meta-analytic frameworks by treating meta-analysis as a prediction problem. It describes neuroscience concepts of interest by continuous combinations of terms rather than matching publications for exact terms. As it combines multiple terms and interpolates between available studies, it extends the scope of meta-analysis in neuroimaging. In particular, it can capture information for concepts studied much less frequently than those that are covered by current automated meta-analytic approaches.

### Related work

A variety of prior works have paved the way for NeuroQuery. Brainmap (*Laird et al., 2005*) was the first systematic database of brain coordinates. NeuroSynth (*Yarkoni et al., 2011*) pioneered automated meta-analysis using abstracts from the literature, broadening a lot the set of terms for which the consistency of reported locations can be tested. These works perform classic meta-analysis, which considers terms in isolation, unlike NeuroQuery. Topic models have also been used to find relationships across terms used in meta-analysis. *Nielsen et al. (2004)* used a non-negative matrix factorization on the matrix of occurrences of terms for each brain location (voxel): their model outputs a set of seven spatial networks associated with cognitive topics, described as weighted combinations of terms. *Poldrack et al. (2012)* used topic models on the full text of 5800 publications to extract from term cooccurrences 130 topics on mental function and disorders, followed by a classic meta-analysis to map their neural correlates in the literature. These topic-modeling works produce a reduced number of cognitive latent factors –or topics– mapped to the brain, unlike NeuroQuery which strives to map *individual* terms and uses their cooccurences in publications only to infer the semantic links. From a modeling perspective, the important difference of NeuroQuery is supervised learning, used as an encoding model (*Naselaris et al., 2011*). In this sense, the supervised learning used in NeuroQuery differs from that used in *Yarkoni et al. (2011)* : the latter is a decoding model that, given brain locations in a study, predicts the likelihood of neuroscience terms without using relationships between terms. Unlike prior approaches, the maps of NeuroQuery are predictions of its statistical model, as opposed to model parameters. Finally, other works have modelled co-activations and interactions between brain locations (*Kang et al., 2011*; *Wager et al., 2015*; *Xue et al.,*

*2014*). We do not explore this possibility here, and except for the density estimation NeuroQuery treats voxels independently.

## Usage recommendations and limitations

We have thoroughly validated that NeuroQuery gives quantitatively and qualitatively good results that summarize well the literature. Yet, the tool has strengths and weaknesses that should inform its usage. Brain maps produced by NeuroQuery are predictions, and a specific prediction may be wrong although the tool performs well on average. A NeuroQuery prediction by itself therefore does not support definite conclusions as it does not come with a statistical test. Rather, NeuroQuery will be most successfully used to produce hypotheses and as an exploratory tool, to be confronted with other sources of evidence. To prepare a new functional neuroimaging study, NeuroQuery helps to formulate hypotheses, defining ROIs or other formal priors (for Bayesian analyses). To interpret results of a neuroimaging experiment, NeuroQuery can readily use the description of the experiment to assemble maps from the literature, which can be compared against, or updated using, experimental findings. As an exploratory tool, extracting patterns from published neuroimaging findings can help conjecture relationships across mental processes as well as their neural correlates (*Yeo et al., 2015*). NeuroQuery can also facilitate literature reviews: given a query, it uses its semantic model to list related studies and their reported activations. What NeuroQuery does not do is provide conclusive evidence that a brain region is recruited by a mental process or affected by a pathology. Compared to traditional meta-analysis tools, NeuroQuery is particularly beneficial (*i*) when the term of interest is rare, (*ii*) when the concept of interest is best described by a combination of multiple terms, and (*iii*) when a fully automated method is necessary and queries would otherwise need cumbersome manual curation to be understood by other tools.

Understanding the components of NeuroQuery helps interpreting its results. We now describe in details potential failures of the tool, and how to detect them. NeuroQuery builds predictions by combining brain maps each associated with a keyword related to the query. A first step to interpret results is to inspect this list of keywords, displayed by the online tool. These keywords are selected based on their semantic relation to the query, and as such will usually be relevant. However, in rare cases, they may build upon undesirable associations. For example, 'agnosia' is linked to 'visual', 'fusiform', 'word' and 'object', because visual agnosia is the type of agnosia most studied in the literature, even though 'agnosia' is a much more general concept. In this specific case, the indirect association is problematic because 'agnosia' is not a *selected* term that NeuroQuery can map by itself, as it is not well-represented in the source data. As a result, the NeuroQuery prediction for 'agnosia' is driven by indirect associations, and focuses on the visual system, rather than areas related to, for example auditory agnosia. By contrast, 'aphasia' is an example of a term that is well mapped, building on maps for 'speech' and 'language', terms that are semantically close to aphasia and well captured in the literature.

A second consideration is that, in some extreme cases, the semantic smoothing fails to produce meaningful results. This happens when a term has no closely related terms that correlate well with brain activity. For instance, 'ADHD' is very similar to 'attention deficit hyperactivity disorder', 'hyperactivity', 'inattention', but none of these terms is selected as a feature mapped in itself, because their link with brain activity is relatively loose. Hence, for 'ADHD', the model builds its prediction on terms that are distant from the query, and produces a misleading map that highlights mostly the cerebellum (https://neuroquery.org/query?text=adhd). While this result is not satisfying, the failure is detected by the NeuroQuery interface and reported with a warning stating that results may not be reliable. To a user with general knowledge in psychology, the failure can also be seen by inspecting the associated terms, as displayed in the user interface.

A third source of potential failure stems from NeuroQuery's model of additive combination. This model is not unique to NeuroQuery, and lies at the heart of functional neuroimaging, which builds upon the hypothesis of pure insertion of cognitive processes (*Ulrich et al., 1999*; *Poldrack, 2010*). An inevitable consequence is that, in some cases, a group of words will not be well mapped by its constituents. For example, 'visual sentence comprehension' is decomposed into two constituents known to Neuroquery: 'visual' and 'sentence comprehension'. Unfortunately, the map corresponding to the combination is then dominated by the primary visual cortex, given that it leads to very powerful activations in fMRI. Note that 'visual word comprehension', a slightly more common subject of interest, is decomposed into 'visual word'

and 'comprehension', which leads to a more plausible map, with strong loadings in the visual word form area.

A careful user can check that each constituent of a query is associated with a plausible map, and that they are well combined. The NeuroQuery interface enables to gauge the quality of the mapping of each individual term by presenting the corresponding brain map as well as the number of associated studies. The final combination can be understood by inspecting the weights of the combination as well as comparing the final combined map with the maps for individual terms. Such an inspection can for instance reveal that, as mentioned above, 'visual' dominates 'sentence comprehension' when mapping 'visual sentence comprehension'.

We have attempted to provide a comprehensive overview of the main pitfalls users are likely to encounter when using NeuroQuery, but we hasten to emphasize that all of these pitfalls are infrequent. NeuroQuery produces reliable maps for the typical queries, as quantified by our experiments.

## General considerations on meta-analyses

When using NeuroQuery to foster scientific progress, it is useful to keep in mind that meta-analyses are not a silver bullet. First, meta-analyses have little or no ability to correct biases present in the primary literature (e.g., perhaps confirmation bias drives researchers to overreport amygdala activation in emotion studies). Beyond increased statistical power, one promise of meta-analysis is to afford a wider perspective on results—in particular, by comparing brain structures detected across many different conditions. However, claims that a structure is selective to a mental condition need an explicit statistical model of reverse inference (*Wager et al., 2016*). Gathering such evidence is challenging: selectivity means that changes at the given brain location specifically *imply* a mental condition, while brain imaging experiments most often do not manipulate the brain itself, but rather the experimental conditions it is placed in *Poldrack (2006)*. In a meta-analysis, the most important confound for reverse inferences is that some brain locations are reported for many different conditions. NeuroQuery accounts for this varying baseline across the brain by fitting an intercept and reporting only differences from the baseline. While helpful, this is not a formal statistical test of reverse inference. For example, the NeuroQuery map for 'interoception' highlights the insula, because studies that mention 'interoception' tend to mention and report coordinates in the insula. This, of course, does *not* mean that interoception is the only function of the insula. Another fundamental challenge of meta-analyses in psychology is the decomposition of the tasks in mental processes: the descriptions of the dimensions of the experimental paradigms are likely imperfect and incomplete. Indeed, even for a task as simple as finger tapping, minor variations in task design lead to reproducible variations in neural responses (*Witt et al., 2008*). However, quantitatively describing all aspects of all tasks and cognitive strategies is presently impossible, as it would require a universally-accepted, all-encompassing psychological ontology. Rather, NeuroQuery grounds meta-analysis in the full-text descriptions of the studies, which in our view provide the best available proxy for such an idealized ontology.

## Conclusion

NeuroQuery stems from a desire to compile results across studies and laboratories, an essential endeavor for the progress of human brain mapping (*Yarkoni et al., 2010*). Mental processes are difficult to isolate and findings of individual studies may not generalize. Thus, tools are needed to denoise and summarize knowledge accumulated across a large number of studies. Such tools must be usable in practice and match the needs of researchers who exploit them to study human brain function and disorders. NeuroSynth took a huge step in this direction by enabling anyone to perform, in a few seconds, a fully automated meta-analysis across thousands of studies, for an important number of isolated terms. Still, users are faced with the difficult task of mapping their question to a single term from the NeuroSynth vocabulary, which cannot always be done in a meaningful way. If the selected term is not popular enough, the resulting map also risks being unusable for lack of statistical power. NeuroQuery provides statistical maps for arbitrary queries – from seldom-studied terms to free-text descriptions of experimental protocols. Thus, it enables applying fully-automated and quantitative meta-analysis in situations where only semi-manual and subjective solutions were

available. It therefore brings an important advancement towards grounding neuroscience on quantitative knowledge representations.

# Materials and methods

We now expose methodological details: first the constitution of the NeuroQuery data, then the statistical model, the validation experiments in details, and the word-occurrence statistics in the corpus of studies.

## Building the NeuroQuery training data

### A new dataset

The dataset collected by NeuroSynth (*Yarkoni et al., 2011*) is openly available (https://github.com/neurosynth/neurosynth-data; copy archived at https://github.com/elifesciences-publications/neurosynth-data). In July, 2019, NeuroSynth contains 448255 unique locations for 14371 studies. It also contains the term frequencies for 3228 terms (1335 are actually used in the NeuroSynth online tool (http://neurosynth.org), based on the abstracts of the studies. However, it only contains term frequencies for the abstracts, and not the articles themselves. This results in a shallow description of the studies, based on a very short text (around 20 times smaller than the full article). As a result, many important terms are very rare: they seldom occur in abstracts, and can be associated with very few studies. For example, in our corpus of 13459 studies, 'huntington disease' occurs in 32 abstracts, and 'prosopagnosia' in 25. For such terms, meta-analysis lacks statistical power. When the full text is available, many more term occurrences – associations between a term and a study – are observed (Figure 16). This means that more information is available, terms are better described by their set of associated studies, and meta-analyses have more statistical power. Moreover, as publications cannot always be redistributed for copyright reasons, NeuroSynth (and any dataset of this nature) can only provide term frequencies for a fixed vocabulary, and not the text they were extracted from. We therefore decided to collect a new corpus of neuroimaging studies, which contains the full text. We also created a new peak activation coordinate extraction system, which achieved a higher precision and recall than NeuroSynth's on a small sample of manually annotated studies.

### Journal articles in a uniform and validated format

We downloaded around 149000 full-text journal articles related to neuroimaging from the PubMed Central (https://www.ncbi.nlm.nih.gov/pmc/, https://www.ncbi.nlm.nih.gov/books/NBK25501/) (*Sayers, 2009*) and Elsevier (https://dev.elsevier.com/api_docs.html) APIs. We focus on these sources of data because they provide many articles in a structured format. It should be noted that this could result in a selection bias, as some scientific journals – mostly paid journals – are not available through these channels. The articles are selected by querying the ESearch Entrez utility (*Sayers, 2009*) either for specific neuroimaging journals or with query strings such as 'fMRI'. The resulting studies are mostly based on fMRI experiments, but the dataset also contains Positron Emission Tomography (PET) or structural Magnetic Resonance Imaging (MRI) studies. It contains studies about diverse types of populations: healthy adults, patients, elderly, children.

We use eXtensible Stylesheet Language Transformations (XSLT) to convert all articles to the Journal Article Tag Suite (JATS) Archiving and Interchange XML language (https://jats.nlm.nih.gov/archiving/) and validate the result using the W3C XML Schema (XSD) schemas provided on the JATS website. From the resulting XML documents, it is straightforward to extract the title, keywords, abstract, and the relevant parts of the article body, discarding the parts which would add noise to our data (such as the acknowledgements or references).

### Coordinate extraction

We extract tables from the downloaded articles and convert them to the XHTML 1.1 table model (the JATS also allows using the OASIS CALS table model). We use stylesheets provided by docbook (https://docbook.org/tools/) to convert from CALS to XHTML. Cells in tables can span several rows and columns. When extracting a table, we normalize it by splitting cells that

span several rows or columns and duplicating these cells' content; the normalized table thus has the shape of a matrix. Finally, all unicode characters that can be used to represent '+' or '-' signs (such as − 'MINUS SIGN') are mapped to their ASCII equivalents, '+' (+ 'PLUS SIGN') or '-" (- 'HYPHEN MINUS'). Once tables are isolated, in XHTML format, and their rows and columns are well aligned, the last step is to find and extract peak activation coordinates. Heuristics find columns containing either single coordinates or triplets of coordinates based on their header and the cells' content. A heuristic detects when the coordinates extracted from a table are probably not stereotactic peak activation coordinates, either because many of them lie outside a standard brain mask, or because the group of coordinates as a whole fits a normal distribution too well. In such cases the whole table is discarded. Out of the 149000 downloaded and formatted articles, 13459 contain coordinates that could be extracted by this process, resulting in a total of 418772 locations.

All the extracted coordinates are treated as coordinates in the Montreal Neurological Institute (MNI) space, even though some articles still refer to the Talairach space. The precision of extracted coordinates could be improved by detecting which reference is used and transforming Talairach coordinates to MNI coordinates. However, differences between the two coordinate systems are at most of the order of 1 cm, and much smaller in most of the brain. This is comparable to the size of the Gaussian kernel used to smooth images. Moreover, the alignment of brain images does not only depend on the used template but also on the registration method, and there is no perfect transformation from Talairach to MNI space (*Lancaster et al., 2007*). Therefore, treating all coordinates uniformly is acceptable as a first approximation, but better handling of Talairach coordinates is a clear direction for improving the NeuroQuery dataset.

## Coordinate extraction evaluation

To evaluate the coordinate extraction process, we focused on articles that are present in both NeuroSynth's dataset and NeuroQuery's, and for which the two coordinate extraction systems disagree. Out of 8692 articles in the intersection of both corpora, the extracted coordinates differ (for at least one coordinate) in 1961 (i.e. in 23% of articles). We selected the first 40 articles (sorted by PubMed ID) and manually evaluated the extracted coordinates. As shown in *Table 2*, our method extracted false coordinates from fewer articles: 3/40 articles have at least one false location in our dataset, against 20 for NeuroSynth. While these numbers may seem high, note that errors are far less likely to occur in articles for which both methods extract exactly the same locations.

## Density maps

For each article, the coordinates from all tables are pooled, resulting in a set of peak activation coordinates. We then use Gaussian Kernel Density Estimation (KDE) (*Silverman, 1986*; *Scott, 2015*) to estimate the density of these activations over the brain. The chosen bandwidth of the Gaussian kernel yields a Full Width at Half Maximum (FWHM) close to 9 mm, which is in the range of smoothing kernels that are typically used for fMRI meta-analysis (*Wager et al., 2007*; *Wager et al., 2004*; *Turkeltaub et al., 2002*). For comparison, NeuroSynth uses a hard ball of 10 mm radius.

One benefit of focusing on the density of peak coordinates (which is $\ell_1$-normalized) is that it does not depend on the number of contrasts presented in an article, nor on other analytic choices that

**Table 2.** Number of extracted coordinate sets that contain at least one error of each type, out of 40 manually annotated articles.

The articles are chosen from those on which NeuroSynth and NeuroQuery disagree – the ones most likely to contain errors.

|  | False positives | False negatives |
| --- | --- | --- |
| NeuroSynth | 20 | 28 |
| NeuroQuery | 3 | 8 |

cause the number of reported coordinates to vary widely, ranging from less than a dozen to several hundreds.

## Vocabulary and TFIDF features

We represent the text of our articles by TFIDF features (*Salton and Buckley, 1988*). These simple representations are popular in document retrieval and text classification because they are very efficient for many applications. They contain the (reweighted) frequencies of many terms in the text, discarding the order in which words appear. An important choice when building TFIDF vectors is the vocabulary: the words or expressions whose frequency are measured. It is common to use all words encountered in the training corpus, possibly discarding those that are too frequent or too rare. The vocabulary is often enriched with 'n-grams', or collocations: groups of words that often appear in the same sequence, such as 'European Union' or 'default mode network'. These collocations are assigned a dimension of the TFIDF representations and counted as if they were a single token. There are several strategies to discover such collocations in a training corpus (*Mikolov et al., 2013*; *Bouma, 2009*).

We do not extract the vocabulary and collocations from the training corpus, but instead rely on existing, manually-curated vocabularies and ontologies of neuroscience. This ensures that we only consider terms that are relevant to brain function, anatomy or disorders, and that we only use meaningful collocations. Moreover, it helps to reduce the dimensionality of the TFIDF representations. Our vocabulary comprises five important lexicons of neuroscience, based on community efforts: the subset of Medical Subject Headings (MeSH) (https://www.ncbi.nlm.nih.gov/mesh) dedicated to neuroscience and psychology, detailed in Section 'The choice of vocabulary' (MeSH are the terms used by PubMed to index articles), Cognitive Atlas (http://www.cognitiveatlas.org/), NeuroNames (http://braininfo.rprc.washington.edu/NeuroNames.xml) and NIF (https://neuinfo.org/). We also include all the terms and bigrams used by NeuroSynth (http://neurosynth.org). We discard all the terms and expressions that occur in less than 5/10 000 articles. The resulting vocabulary contains 7547 terms and expressions related to neuroscience.

## Summary of collected data

The data collection described in this section provides us with important resources: (*i*) Over 149K full-text journal articles related to neuroscience – 13.5K of which contain peak activation coordinates – all translated into the same structured format and validated. (*ii*) Over 418K peak activation coordinates for more than 13.5K articles. (*iii*) A vocabulary of 7547 terms related to neuroscience, each occurring in at least six articles from which we extracted coordinates. This dataset is the largest of its kind. In what follows we focus on the set of 13.5K articles from which we extracted peak locations.

Some quantitative aspects of the NeuroQuery and NeuroSynth datasets are summarized in *Table 3*.

### Text

In terms of raw amount of text, this corpus is 20 times larger than NeuroSynth's. Combined with our vocabulary, it yields over 5.5M occurrences of a unique term in an article. This is over five times more than the word occurrence counts distributed by NeuroSynth (https://github.com/neurosynth/neurosynth-data). When considering only terms in NeuroSynth's vocabulary, the corpus still contains over 3M term-study associations, 4.6 times more than NeuroSynth. Using this larger corpus results in denser representations, higher statistical power, and coverage of a wider vocabulary. There is an important overlap between the selected studies: 8 692 studies are present in both datasets – the Intersection Over Union is 0.45.

### Coordinates

The set of extracted coordinates is almost the size of NeuroSynth's (which is 7% larger with 448255 coordinates after removing duplicates), and is less noisy. To compare coordinate extractions, we manually annotated a small set of articles for which NeuroSynth's coordinates differ from NeuroQuery's. Compared with NeuroSynth, NeuroQuery's extraction method reduced the number of articles with incorrect coordinates (false positives) by a factor of 7, and the number of articles with

**Table 3.** Comparison with NeuroSynth.

'voc intersection' is the set of terms present in both NeuroSynth's and NeuroQuery's vocabularies. The 'conflicting articles' are papers present in both datasets, for which the coordinate extraction tools disagree, 40 of which were manually annotated.

| | NeuroSynth | NeuroQuery |
|---|---|---|
| Dataset size | | |
| articles | 14 371 | 13 459 |
| terms | 3 228 (1 335 online) | 7 547 |
| journals | 60 | 458 |
| raw text length (words) | ≈4 M | ≈75 M |
| unique term occurrences | 1 063 670 | 5 855 483 |
| unique term occurrences in voc intersection | 677 345 | 3 089 040 |
| coordinates | 448 255 | 418 772 |
| Coordinate extraction errors on conflicting articles | | |
| articles with false positives / 40 | 20 | 3 |
| articles with false negatives / 40 | 28 | 8 |

missing coordinates (false negatives) by a factor of 3 (*Table 2*). Less noisy brain activation data is useful for training encoding models.

## Sharing data

We do not have the right to share the full text of the articles, but the vocabulary, extracted coordinates, and term occurrence counts for the whole corpus are freely available online (https://github.com/neuroquery/neuroquery_data).

## Mathematical details of the NeuroQuery statistical model

### Notation

We denote scalars, vectors and matrices with lower-case, bold lower-case, and bold-upper case letters respectively: $x$, $\boldsymbol{x}$, $\boldsymbol{X}$. We denote the elements of $\boldsymbol{X}$ by $x_{i,j}$, its rows by $\boldsymbol{x}_i$, and its columns by $\boldsymbol{x}_{*,i}$. We denote $p$ the number of voxels in the brain, $v$ the size of the vocabulary, and $n$ the number of studies in the dataset. We use indices $i$, $j$, $k$ to indicate indexing samples (studies), features (terms), and outputs (voxels) respectively. We use a hat to denote estimated values, for example $\hat{\boldsymbol{B}}$. $\langle \boldsymbol{x}, \boldsymbol{y} \rangle$ is the vector scalar product.

### TFIDF feature extraction

We represent a document by its TFIDF features (*Salton and Buckley, 1988*), which are reweighted Bag-Of-Words features. A TFIDF representation is a vector in which each entry corresponds to the (reweighted) frequency of occurrence of a particular term. The *term frequency*, tf, of a word in a document is the number of times the word occurs, divided by the total number of words in the document. The *document frequency*, df, of a word in a corpus is the proportion of documents in which it appears. The *inverse document frequency*, idf, is defined as:

$$\text{idf}(w) = -\log(*df) + 1 = -\log\frac{|\{i \,|\, w \text{ occurs in document i}\}|}{n} + 1\,, \tag{1}$$

where $n$ is the number of documents in the corpus and $|\cdot|$ is the cardinality. Term frequencies are reweighted by their idf, so that frequent words, which occur in many documents (such as 'results' or 'brain'), are given less importance. Indeed, such words are usually not very informative.

**Table 4.** Atlases included in NeuroQuery's vocabulary.

| Name | Url |
| --- | --- |
| talairach | http://www.talairach.org/talairach.nii |
| harvard_oxford | http://www.nitrc.org/frs/download.php/7700/HarvardOxford.tgz |
| destrieux | https://www.nitrc.org/frs/download.php/7739/destrieux2009.tgz |
| aal | http://www.gin.cnrs.fr/AAL-217 |
| JHU-labels | https://fsl.fmrib.ox.ac.uk/fsl/fslwiki/Atlases#JHU-labels |
| Striatum-Structural | https://fsl.fmrib.ox.ac.uk/fsl/fslwiki/Atlases#Striatum-Structural |
| STN | https://fsl.fmrib.ox.ac.uk/fsl/fslwiki/Atlases#STN |
| Striatum-Connectivity-7sub | https://fsl.fmrib.ox.ac.uk/fsl/fslwiki/Atlases#Striatum-Connectivity-7sub |
| Juelich | https://fsl.fmrib.ox.ac.uk/fsl/fslwiki/Atlases#Juelich |
| MNI | https://fsl.fmrib.ox.ac.uk/fsl/fslwiki/Atlases#MNI |
| JHU-tracts | https://fsl.fmrib.ox.ac.uk/fsl/fslwiki/Atlases#JHU-tracts |
| Thalamus | https://fsl.fmrib.ox.ac.uk/fsl/fslwiki/Atlases#Thalamus |

Our TFIDF representation for a study is the uniform average of the normalized TFIDF vectors for its title, abstract, full text, and keywords. Therefore, all parts of the article are taken into account, but a word that occurs in the title is more important than a word the article body (since the title is shorter).

TFIDF features exploit a fixed vocabulary – each dimension is associated with a particular word. The vocabulary we consider comprises 7547 terms or phrases related to neuroscience that occur in at least 0.05% of publications. These terms are extracted from manually curated sources shown in *Table 1* and *Table 4*.

## Reweighted ridge matrix and feature (vocabulary) selection

Here we give some details about the feature selection and adaptive ridge regularization. After extracting TFIDF features and computing density estimation maps, we fit a linear model by regressing the activity of each voxel on the TFIDF descriptors (Section 'Overview of the NeuroQuery model'). We denote $p$ the number of voxels, $v$ the size of the vocabulary, and $n$ the number of documents in the corpus. We construct a design matrix $X \in \mathbb{R}^{n \times v}$ containing the TFIDF features of each study, and the dependent variables $Y \in \mathbb{R}^{n \times p}$ representing the activation density at each voxel for each study. The linear model thus writes:

$$Y = X B^* + E, \tag{2}$$

where $E$ is Gaussian noise and $B^* \in \mathbb{R}^{v \times p}$ are the unknown model coefficients. We use ridge regression (least-squares regression with a penalty on the $\ell_2$ norm of the model coefficients). Some words are much more informative than others, or have a much stronger correlation with brain activity. For example, 'auditory' is well correlated with activations in the auditory areas, whereas 'attention' has a lower signal-to-noise ratio, as it is polysemic and, even when used as a psychological concept, has a weaker link to reported neural activations. Therefore it is beneficial to adapt the amount of regularization for each word, to strongly penalize (or even discard) the most noisy features.

Many existing methods for feature selection are not adapted to our case, because: (*i*) the design matrix $X$ is very sparse, and more importantly (*ii*) we want to select the same features for $\approx$ 28 000 outputs (each voxel in the brain is a dependent variable). We therefore introduce a new reweighted ridge regression and feature selection procedure.

Our approach is based on the observation that when fitting a ridge regression with a uniform regularization, the most informative words are associated with large coefficients for many voxels. We start by fitting a ridge regression with uniform regularization. We obtain one statistical map of the brain for every feature (every term in the vocabulary). The maps are rescaled to reduce the

importance of coefficients with a high variance. We then compute the squared $\ell_2$ norms of these brain maps across voxels. These norms are a good proxy for the importance of each feature. Terms associated with large norms explain well the activity of many voxels and tend to be helpful features. We rely on these brain map norms to determine which features are selected and what regularization is applied. The feature selection and adaptive regularization are described in detail in the rest of this section.

### Z scores for ridge regression coefficients

Our design matrix $X \in \mathbb{R}^{n \times v}$ holds TFIDF features for $v$ terms in $n$ studies. There are $p$ dependent variables, one for each voxel in the brain, which form $Y \in \mathbb{R}^{n \times p}$. The first ridge regression fit yields coefficients $\hat{B}^{(0)} \in \mathbb{R}^{v \times p}$:

$$\hat{B}^{(0)} = \underset{B \in \mathbb{R}^{v \times p}}{\operatorname{argmin}} ||Y - XB||_{*F}^2 + \lambda ||B||_{*F}^2, \tag{3}$$

where $\lambda \in \mathbb{R}_{>0}$ is a hyperparameter set with Generalized Cross-Validation (GCV); (*Rifkin and Lippert, 2007*). We then compute an estimate of the variance of these coefficients. The approach is similar to the one presented in *Gaonkar and Davatzikos (2012)* for the case of SVMs. A simple estimator can be obtained by noting that the coefficients of a ridge regression are a linear function of the dependent variables. Indeed, solving *Equation 3* yields:

$$\hat{B}^{(0)} = (X^T X + \lambda I)^{-1} X^T Y. \tag{4}$$

Defining

$$M = (X^T X + \lambda I)^{-1} X^T \in \mathbb{R}^{v \times n}, \tag{5}$$

for a voxel $k \in \{1, \ldots, p\}$, and a feature $j \in \{1, \ldots v\}$,

$$\hat{b}_{j,k}^{(0)} = \langle m_j, y_{*,k} \rangle, \tag{6}$$

where $m_j \in \mathbb{R}^n$ is the $i^{\text{th}}$ row of $M$ and $y_{*,k} \in \mathbb{R}^n$ is the $k^{\text{th}}$ column of $Y$. The activations of voxel $k$ across studies are considered to be independent identically distributed (i.i.d), so

$$\operatorname{Var}(y_{*,k}) = *\operatorname{Var}(y_{1,k}) I_n \overset{\triangle}{=} s_k^2 I_n. \tag{7}$$

An estimate of this variance can be obtained from the residuals:

$$\hat{s}_k^2 \overset{\triangle}{=} \frac{1}{n} \sum_{i=1}^n (\hat{y}_{i,k}^{(0)} - y_{i,k})^2 = \frac{1}{n} \sum_{i=1}^n ((X\hat{B}^{(0)})_{i,k} - y_{i,k})^2. \tag{8}$$

A simple estimate of the coefficients' variance is then:

$$\hat{\sigma}_{j,k}^2 \overset{\triangle}{=} *\widehat{Var}(\hat{b}_{j,k}^{(0)}) = \hat{s}_k^2 \langle m_j, m_j \rangle = \hat{s}_k^2 \sum_{i=1}^n m_{j,i}^2 \tag{9}$$

We can thus estimate the standard deviation of each entry of $\hat{B}^{(0)}$. We obtain a brain map of Z scores for each term in the vocabulary: for term $j \in \{1, \ldots, v\}$ and voxel $k \in \{1 \ldots p\}$,

$$\hat{z}_{j,k} \overset{\triangle}{=} \frac{\hat{b}_{j,k}^{(0)}}{\hat{\sigma}_{j,k}}. \tag{10}$$

We denote $\hat{\sigma}_j = (\hat{\sigma}_{j,1}, \ldots, \hat{\sigma}_{j,p}) \in \mathbb{R}^p$; and the Z-map for term $j$.

### Reweighted ridge matrix

Once we have a Z-map for each term, we summarize these maps by computing their squared Euclidean norm. In practice, we smooth the Z scores: $\hat{z}_{j,k}$ in *Equation 10* is replaced by

$$\hat{\zeta}_{j,k} = \frac{\hat{b}_{j,k}^{(0)}}{\hat{\sigma}_{j,k} + \delta}, \tag{11}$$

where $\delta$ is a constant offset. The offset $\delta$ allows us to interpolate between basing the regularization on the Z scores, or on the raw coefficients, that is the $\beta$-maps. We obtain better results with a large value for $\delta$, such as the mean variance of all the regression coefficients. This prevents selecting terms only because they have a very small estimated variance in some voxels. Note that this offset $\delta$ is only used to compute the regularization, and not to compute the rescaled predictions produced by NeuroQuery as in *Equation 17*.

We denote $\hat{\zeta}_j = (\hat{\zeta}_{j,1}, \ldots, \hat{\zeta}_{j,p}) \in \mathbb{R}^p$, $\forall j \in \{1, \ldots, v\}$. Next, we compute the mean $\mu$ and standard deviation $e$ of $\{ ||\hat{\zeta}_j||_2^2, j = 1 \ldots v \}$, and set an arbitrary cutoff

$$c = \mu + 2e. \tag{12}$$

All features $j$ such that $||\hat{\zeta}_j||_2^2 \leq c + \epsilon$, where $\epsilon$ is a small margin to avoid division by zero in *Equation 14*, are discarded. In practice we set $\epsilon$ to 0.001. The value of $\epsilon$ is not important, because features that are not discarded but have their $\zeta$ norm close to $c$ get very heavily penalized in *Equation 14* and have coefficients very close to .

We denote $u < v$ the number of features that remain in the selected vocabulary. We denote $\phi : \{1 \ldots u\} \to \{1 \ldots v\}$ the strictly increasing mapping that reindexes the features by keeping only the $u$ selected terms: $\phi(\{1 \ldots u\})$ is the set of selected features. We denote $P \in \mathbb{R}^{u \times v}$ the corresponding projection matrix:

$$p_{*,j}^T = e_{\phi(j)}, \forall j \in \{1 \ldots u\}, \tag{13}$$

where $\{e_j, j = 1 \ldots v\}$ is the natural basis of $\mathbb{R}^v$. The regularization for the selected features is then set to

$$w_j = \frac{1}{||\hat{\zeta}_{\phi(j)}||_2^2 - c}. \tag{14}$$

Finally, we define the diagonal matrix $W \in \mathbb{R}^{u \times u}$ such that the $j^{\text{th}}$ element of its diagonal is $w_j$ and fit a new set of coefficients $\hat{B} \in \mathbb{R}^{u \times p}$ with this new ridge matrix.

## Fitting the reweighted ridge regression

The reweighted ridge regression problem writes:

$$\hat{B} = \underset{B \in \mathbb{R}^{u \times p}}{\operatorname{argmin}} ||Y - XP^T B||_{*F}^2 + \gamma * Tr(B^T W B), \tag{15}$$

Where $\gamma \in \mathbb{R}_{>0}$ is a new hyperparameter, that is again set by Generalized Cross-Validation (GCV). With a change of variables this becomes equivalent to solving the usual ridge regression problem:

$$\hat{\Gamma} = \underset{\Gamma}{\operatorname{argmin}} ||Y - \tilde{X}\Gamma||_{*F}^2 + \gamma ||\Gamma||_{*F}^2, \tag{16}$$

where $\tilde{X} = XP^T W^{-\frac{1}{2}}$ and we recover $\hat{B}$ as $\hat{B} = W^{-\frac{1}{2}}\hat{\Gamma}$.

The variance of the parameters $\hat{B}$ can be estimated as in *Equation 9* – without applying the smoothing of *Equation 11*. NeuroQuery can thus report rescaled predictions

$$\hat{z} = \frac{x^T \hat{B}}{\left( *\widehat{Var}(x^T \hat{B}) \right)^{\frac{1}{2}}} \tag{17}$$

One benefit of this rescaling is to provide the user a natural value to threshold the maps. As visible on *Figures 4*, *5* and *6*, thresholding for example at $\hat{z} \approx 3$ selects regions typical of the query, that can be used for instance in a region of interest analysis.

## Summary of the regression with adaptive regularization

The whole procedure for feature selection and adaptive regularization is summarized in Algorithm 1.

---

**Algorithm 1** Reweighted Ridge Regression

---

**Input:** TFIDF features $X$, brain activation densities $Y$, regularization hyperparameter grid $\Lambda$, variance smoothing parameter $\delta$ use

use GCV to compute the best hyperparameter $\lambda \in \Lambda$ and
$\hat{B}^{(0)} = *argmin_B ||Y - XB||^2_{*F} + \lambda ||B||^2_{*F}$;

compute variance estimates $\hat{\sigma}^2_j$ as in **Equation 9**

$\hat{\zeta}_j \leftarrow \frac{\hat{b}^{(0)}_j}{\hat{\sigma}_j + \delta} \forall j \in \{1...v\}$;

compute $c$ according to **Equation 12**

define $\phi$ the reindexing that selects features $j$ such that $||\hat{\zeta}_j||^2_2 > c + \epsilon$;

define $P \in \mathbb{R}^{u \times v}$ the projection matrix for $\phi$ as in **Equation 13**

$w_j \leftarrow \frac{1}{||\hat{\zeta}_{\phi(j)}||^2_2 - c} \forall j \in \{1...u\}$;

$W \leftarrow \mathrm{diag}(w_j, j = 1...u)$;

use GCV to compute the best hyperparameter $\gamma \in \Lambda$ and
$\hat{B} = \underset{B}{\mathrm{argmin}} ||Y - XP^T B||^2_{*F} + \gamma * Tr(B^T WB)$

return $\hat{B}$, $*\widehat{Var}(\hat{B})$, $\gamma$, $P$, $W$

---

In practice, the feature selection keeps u ≈ 200 features. It has a very low computational cost compared to other feature selection schemes. The computational cost is that of fitting two ridge regressions (and the second one is fitted with a much smaller number of features). Moreover, the feature selection also reduces computation at prediction time, which is useful because we deploy an online tool based on the NeuroQuery model (https://neuroquery.org).

## Smoothing: regularization at test time

In order to smooth the sparse input features, we exploit the covariance of our training corpus. We rely on Non-negative Matrix Factorization (NMF) (*Lee and Seung, 1999*). We use a NMF of $X \in \mathbb{R}^{n \times v}$ to compute a low-rank approximation of the covariance $X^T X \in \mathbb{R}^{v \times v}$. Thus, we obtain a denoised term co-occurrence matrix, which measures the strength of association between pairs of terms. We start by computing an approximate factorization of the corpus TFIDF matrix $X$:

$$U, V = \underset{\substack{U \in \mathbb{R}^{n \times d}_{\geq 0} \\ V \in \mathbb{R}^{d \times v}_{\geq 0}}}{\mathrm{argmin}} ||X - UV||^2_{*F} + \lambda (||U||^2_{*F} + ||V||^2_{*F}) + \gamma (||U||_{1,1} + ||V||_{1,1}), \tag{18}$$

where $d < v$ is a hyperparameter and $|| \cdot ||_{1,1}$ designates the sum of absolute values of all entries of a matrix. Computing this factorization amounts to describing each document in the corpus as a linear mixture of $d$ latent factors, or *topics*. In natural language processing, similar decomposition methods are referred to as *topic modelling* (*Blei et al., 2003*; *Deerwester et al., 1990*).

The latent factors, or topics, are the rows of $V \in \mathbb{R}^{d \times v}$: each topic is characterized by a vector of positive weights over the terms in the vocabulary. $U \in \mathbb{R}^{n \times d}$ contains the weight that each document gives to each topic. For each term in the vocabulary, the corresponding column of $V$ is a a $d$-dimensional *embedding* in the low-dimensional, latent space: this embedding contains the strength of association of the term with each topic. These embeddings capture semantic relationships: related terms tend to be associated with embeddings that have large inner products.

The hyperparameters $d = 300$, $\lambda = 0.1$ and $\gamma = 0.01$ are set by evaluating the reconstruction error, sparsity of the similarity matrix, and extracted topics (rows of $V$) on an unlabelled (separate) corpus. We find that the NeuroQuery model as a whole is not very sensitive to these hyperparameters and we obtain similar results for a range of different values.

**Equation 18** is a well-known problem. We solve it with a coordinate-descent algorithm described in *Cichocki and Phan (2009)* and implemented in >scikit-learn (*Pedregosa et al., 2011*). Then, let

$N \in \mathbb{R}^{d \times d}$ be the diagonal matrix containing the Euclidean norms of the columns of $U$, that is such that $n_{ii} = ||\boldsymbol{u}_{*,i}||_2$ and let $\tilde{V} = NV$. We define the word similarity matrix $A = \tilde{V}^T \tilde{V} \in \mathbb{R}^{v \times v}$. This matrix is a denoised, low-rank approximation of the corpus covariance. Indeed

$$X^T X \approx (UV)^T UV \tag{19}$$

$$= V^T N^T (U N^{-1})^T U N^{-1} N V \tag{20}$$

$$\approx \tilde{V}^T \tilde{V} \tag{21}$$

The last approximation is justified by the fact that the columns of $U \in \mathbb{R}^{n \times d}$ are almost orthogonal, and $U^T U$ is almost a diagonal matrix. This is what we observe in practice, and is due to the fact that $n \approx 13\,000$ is much larger than $d = 300$, and that to minimize the reconstruction error in *Equation 18* the columns of $U$ have an incentive to span a large subspace of $\mathbb{R}^n$.

The similarity matrix $A$ contains the inner products of the low-dimensional embeddings of the terms in our vocabulary. We form the matrix $T$ by dividing the rows of $A$ by their $\ell_1$ norm:

$$t_{i,j} = \frac{a_{i,j}}{||a_i||_1} \; \forall i = 1 \ldots v, j = 1 \ldots v. \tag{22}$$

This normalization ensures that terms that have many neighbors are not given more importance in the smoothed representation. The smoothing matrix that we use is then defined as:

$$S = (1 - \alpha) I + \alpha T, \tag{23}$$

with $0 < \alpha < 1$ (in our experiments $\alpha$ is set to 0.1). This smoothing matrix is a mixture of the identity matrix and the term associations $T$. The model is not very sensitive to the parameter $\alpha$ as long as it is chosen small enough for terms actually present in the query to have a higher weight than terms introduced by the query expansion. This prevents degrading performance for documents which contain well-encoded terms, which obtain good prediction even without smoothing. This explains why in *Figure 3*, the prediction for 'visual' relies mostly on the regression coefficient for this exact term, whereas the prediction for 'agnosia' relies on coefficients of terms that are *related* to 'agnosia' – 'agnosia' itself is not kept by the feature selection procedure.

The smoothed representation for a query **q** becomes:

$$x = S^T q \in \mathbb{R}^v \tag{24}$$

where $q \in \mathbb{R}^v$ is the TFIDF representation of the query in large vocabulary space, and $S \in \mathbb{R}^{v \times v}$ is the smoothing matrix. And the prediction for $q$ is:

$$\hat{y} = \hat{B} P S^T q, \tag{25}$$

where $P \in \mathbb{R}^{u \times v}$ is the projection onto the useful vocabulary (selected features), $\hat{B} \in \mathbb{R}^{p \times u}$ are the estimated linear regression coefficients, $\hat{y} \in \mathbb{R}^p$ is the predicted map.

## Validation experiments: additional details

### Example meta-analysis results for the RSVP language task from the IBC dataset

Here we provide more details on the meta-analyses for 'Read pseudo-words vs consonant strings' shown in *Figure 2*. The PMIDS of the studies included in the GingerALE meta-analysis are: 15961322, 16574082, 16968771, 17189619, 17884585, 17933023, 18272399, 18423780, 18476755, 18778780, 19396362, 19591947, 20035884, 20600985, 20650450, 20961169, 21767584, 22285025, 22659111, 23117157, 23270676, 24321558, 24508158, 24667455, 25566039, 26017384, 26188258, 26235228, 28780219. Representing a total of 29 studies and 2025 peak activation coordinates. They are the studies from our corpus (the largest existing corpus of text and peak activation coordinates, with $\approx 14\,000$ studies) which contain the terms: 'reading', 'pseudo', 'word', 'consonant' and 'string'. The map shown on the right of *Figure 2* was obtained with GingerALE, 5000 permutations and the default settings otherwise. Note that an unrealistically low threshold is used for the display because the map would be empty otherwise. *Figure 8* displays more maps with different analysis strategies:

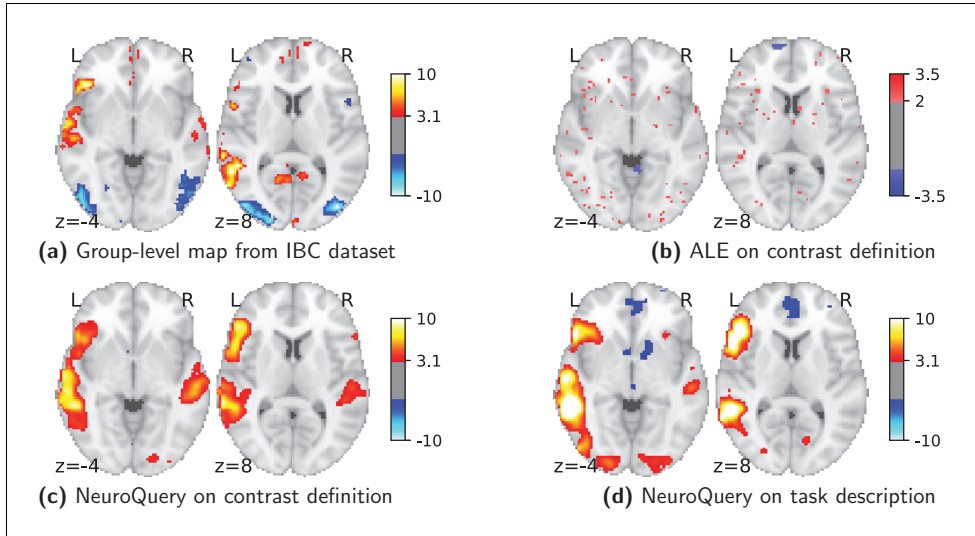

**Figure 8.** Using meta-analysis to interpret fMRI maps. Example of the 'Read pseudo-words vs. consonant strings' contrast, derived from the RSVP language task in the IBC dataset. (**a**): the group-level map obtained from the actual fMRI data from IBC. (**b**): ALE map using the 29 studies in our corpus that contain all five terms from the contrast name. (**c**): NeuroQuery map obtained from the contrast name. (**d**): NeuroQuery map obtained from the page-long RSVP task description in the IBC dataset documentation: https://project.inria.fr/IBC/files/2019/03/documentation.pdf.

the details of the original contrasts and the difference between running NeuroQuery the contrast definition or the task definition. The task definition leads to predicted activations in the early visual cortex, as in the actual group-level maps from the experiment but unlike the predictions from the contrast definition, as the later contains no information on the stimulus modality.

## NeuroQuery performance on unseen pairs of terms

*Figure 4* shows in a qualitative way that NeuroQuery can produce useful brain maps on a combination of terms that have not been studied together. To give a quantitative evaluation that is not limited to a specific pair of terms, we perform a systematic experiment, studying prediction on many unseen pairs of term. For this purpose, we chose pairs of terms in our full corpus and leave out all the studies where both of these terms appear. We train a NeuroQuery model on the reduced corpus of studies obtained by excluding studies with both terms, and evaluate its predictions on the left-out studies.

We choose terms that appear simultaneously in studies frequently (more than 500) to ensure a good estimation of the combined locations for these terms in the test set, but not too frequently (less than 1000), to avoid depleting the training set too much. Indeed, removing the studies for both terms from the corpus not only decreases the statistical power to map these terms but also, more importantly, it creates a negative correlation between these terms. Out of these terms, we select 1000 out random as a left-out and run the experiment 1000 times.

To evaluate NeuroQuery's prediction on these unseen pairs of terms, we first use the same metrics as in Section 'Quantitative evaluation: NeuroQuery is an accurate model of the literature.' *Figure 9* -left shows the log-likelihood of coordinates reported in a publication evaluated on left-out studies that contain the combination of terms excluded from the train set. Compared to testing on a random subset of studied, identically distributed to the training, there is a slight decrease in likelihood but it is small compared to the variance between cross-validation runs. *Figure 9*-right shows results for our other validation metric adapted from *Mitchell et al. (2008)*: matching 1 publication out of 2 to its observed locations. The decrease in performance is more marked. However, it should be noted that the task is more difficult when the test set is made only of publications that all contain two terms, as these publications are all more similar to each other than random publications from the general corpus.

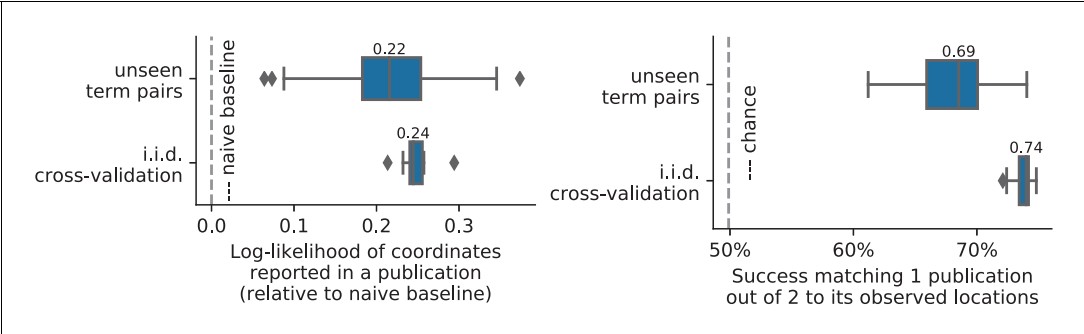

**Figure 9.** Quantitative evaluations on unseen pairs. A quantitative comparison of prediction on random unseen studies (i.i.d.cross-validation) to prediction on studies containing pairs of terms never seen before, using the two measures of predictions performance (as visible on *Figure 7* for standard cross-validation).

To gauge the quality of the maps on unseen pairs, and not only how well the corresponding publications are captured, *Figure 10* shows the Pearson correlation between the predicted brain map and the average density of the reported locations in the left-out studies. The excellent median Pearson correlation of .85 shows that the predicted brain map is indeed true to what a meta-analysis of these studies would reveal.

## NeuroQuery prediction performance without anatomical terms

In *Figure 11*, we present an additional quantitative measure of prediction performance. We delete all terms that are related to anatomy in test articles, to see how NeuroQuery performs without these highly predictive features, which may be missing from queries related to brain function. As the GCLDA and NeuroSynth tools are designed to work with NeuroSynth data, they are only tested on NeuroSynth's TFIDF features, which represent the articles' abstracts.

## Variable terminology

In *Figure 12*, we show predictions for some terms related to mental arithmetic. NeuroQuery's semantic smoothing produces consistent results for related terms.

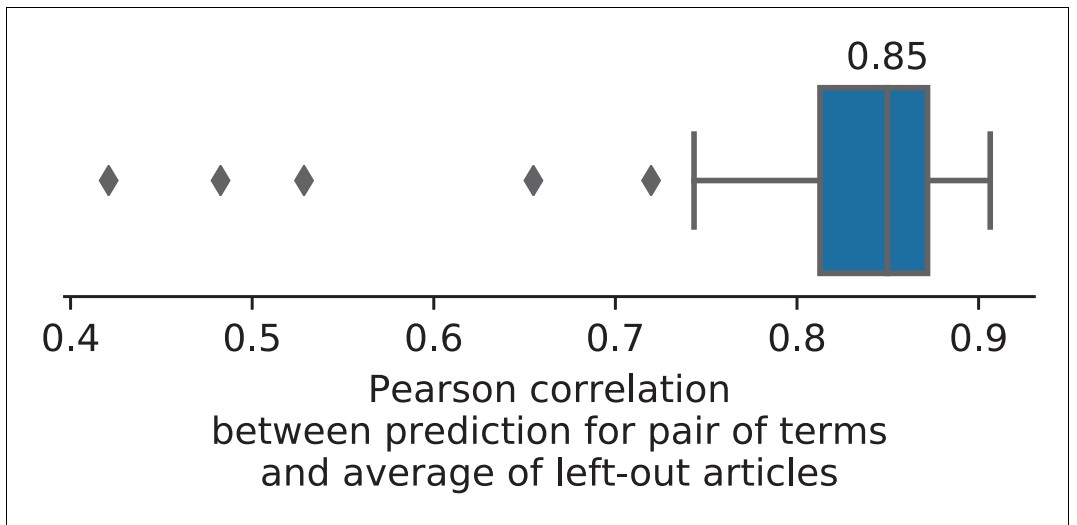

**Figure 10.** Consistency between prediction of unseen pairs and meta-analysis. The Pearson correlation between the map predicted by NeuroQuery on a pair of unseen terms and the average density of locations reported on the studies containing this pair of terms (hence excluded from the training set of NeuroQuery).

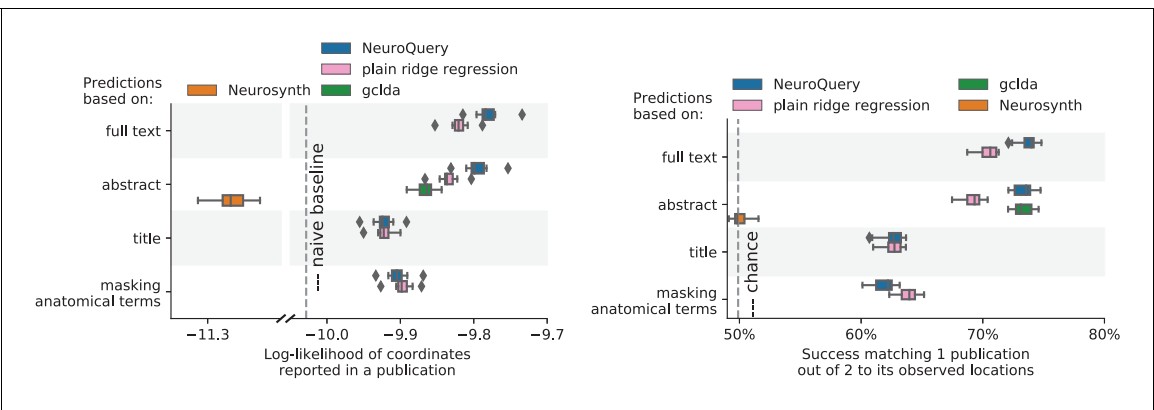

**Figure 11.** Explaining coordinates reported in unseen studies. left: log-likelihood of reported coordinates in test articles. right: how often the predicted map is closer to the true coordinates than to the coordinates for another article in the test set [mitchell2008predicting, ]. The boxes represent the first, second and third quartiles of scores across 16 cross-validation folds. Whiskers represent the rest of the distribution, except for outliers, defined as points beyond 1.5 times the IQR past the low and high quartiles, and represented with diamond fliers.

## Comparison with the BrainPedia IBMA study

To compare maps produced by NeuroQuery with a reliable ground truth, we use the BrainPedia study (*Varoquaux et al., 2018*), which exploits IBMA to produce maps for 19 cognitive concepts. Indeed, when it its feasible, IBMA of manually selected studies produces high-quality brain maps and has been used as a reference for CBMA methods (*Salimi-Khorshidi et al., 2009*). We download the BrainPedia maps and their cognitive labels from the NeuroVault platform (https://neurovault.org/collections/4563/). BrainPedia maps combine forward and reverse inference, and are thresholded to identify regions that are both recruited and predictive of each cognitive process. We treat these maps as a binary ground truth: above-threshold voxels are relevant to the map's label. For each label, we obtain a brain map from NeuroQuery, NeuroSynth and GCLDA. We compare these results to the BrainPedia thresholded maps and measure the Area Under the ROC Curve. This standard classification metric measures the probability that a voxel that is active in the BrainPedia reference map will be given a higher intensity in the NeuroQuery prediction than a voxel that is inactive in the BrainPedia map.

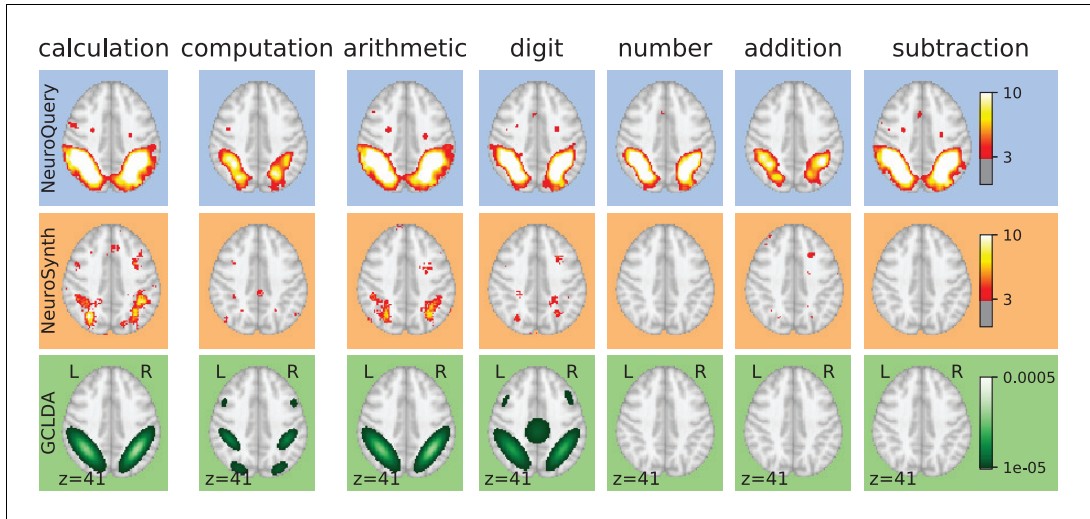

**Figure 12.** Taming arbitrary query variability Maps obtained for a few words related to mental arithmetic. By correctly capturing the fact that these words are related, NeuroQuery can use its map for easier words like 'calculation' and 'arithmetic' to encode terms like 'computation' and 'addition' that are difficult for meta-analysis.

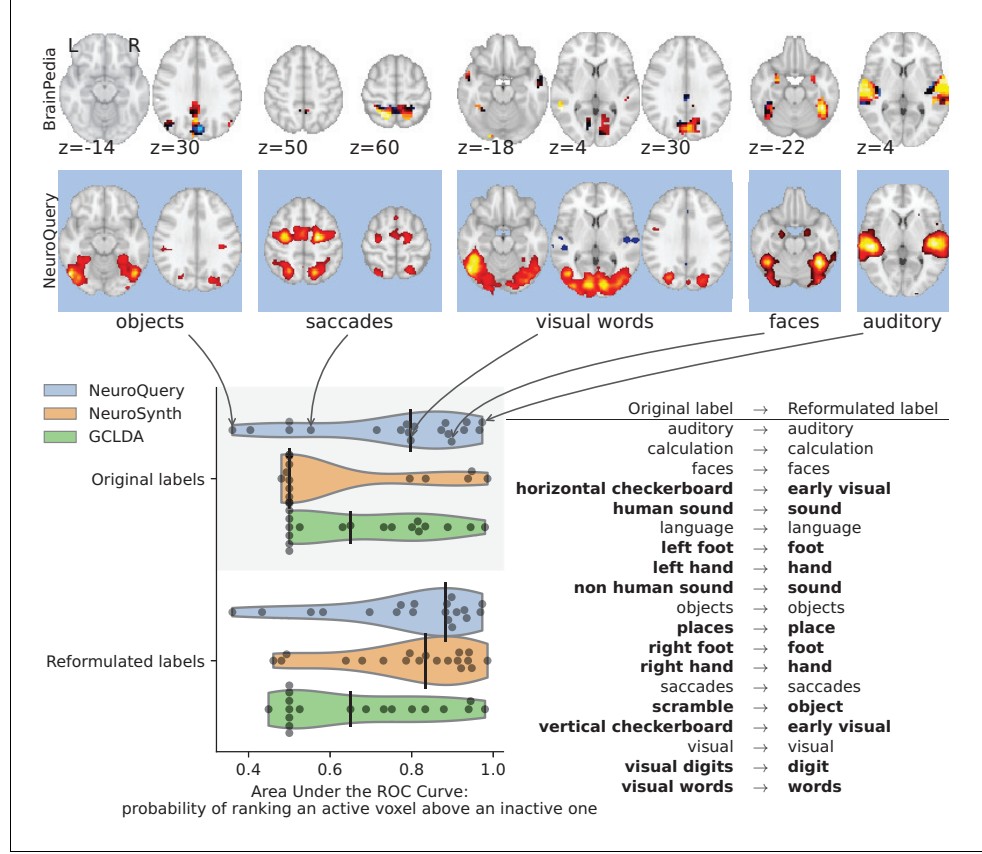

**Figure 13.** Comparison of CBMA maps with IBMA maps from the BrainPedia study. We use labelled and thresholded maps resulting from a manual IBMA. The labels are fed to NeuroQuery, NeuroSynth and GCLDA and their results are compared to the reference by measuring the Area under the ROC Curve. The black vertical bars show the median. When using the original BrainPedia labels, NeuroQuery performs relatively well but NeuroSynth fails to recognize most labels. When reformulating the labels, that is replacing them with similar terms from NeuroSynth's vocabulary, both NeuroSynth and NeuroQuery match the manual IBMA reference for most terms. On the top, we show the BrainPedia map (first row) and NeuroQuery prediction (second row) for the quartiles of the AUC obtained by NeuroQuery on the original labels. A lower AUC for some concepts can sometimes be explained by a more noisy BrainPedia reference map.

We consider two settings. First, we use the original labels provided in the NeuroVault metadata. However, some of these labels are missing from the NeuroSynth vocabulary. In a second experiment, we therefore replace these labels with the most similar term we can find in the NeuroSynth vocabulary. These replacements are shown in *Figure 13*.

When replacing the original labels with less specific terms understood by NeuroSynth, both NeuroQuery and NeuroSynth perform well: NeuroQuery's median AUC is 0.9 and NeuroSynth's is 0.8. When using the original labels, NeuroSynth fails to produce results for many labels as they are missing from its vocabulary. NeuroQuery still performs well on these uncurated labels with a median AUC of 0.8. Finally, we can note that although the BrainPedia maps come from IBMA conducted on carefully selected fMRI studies, they also contain some noise. As can be seen in *Figure 13*, BrainPedia maps that qualitatively match the domain knowledge also tend to be close to the CBMA results produced by NeuroQuery and NeuroSynth.

## Comparison with Harvard-Oxford anatomical atlas

Here, we compare CBMA maps to manually segmented regions of the Harvard-Oxford anatomical atlas (*Desikan et al., 2006*). We feed the labels from this atlas to NeuroQuery, NeuroSynth and GCLDA and compare the resulting maps to the atlas regions. This experiment provides a sanity check that relies on an excellent ground truth, as the atlas regions are labelled and segmented by

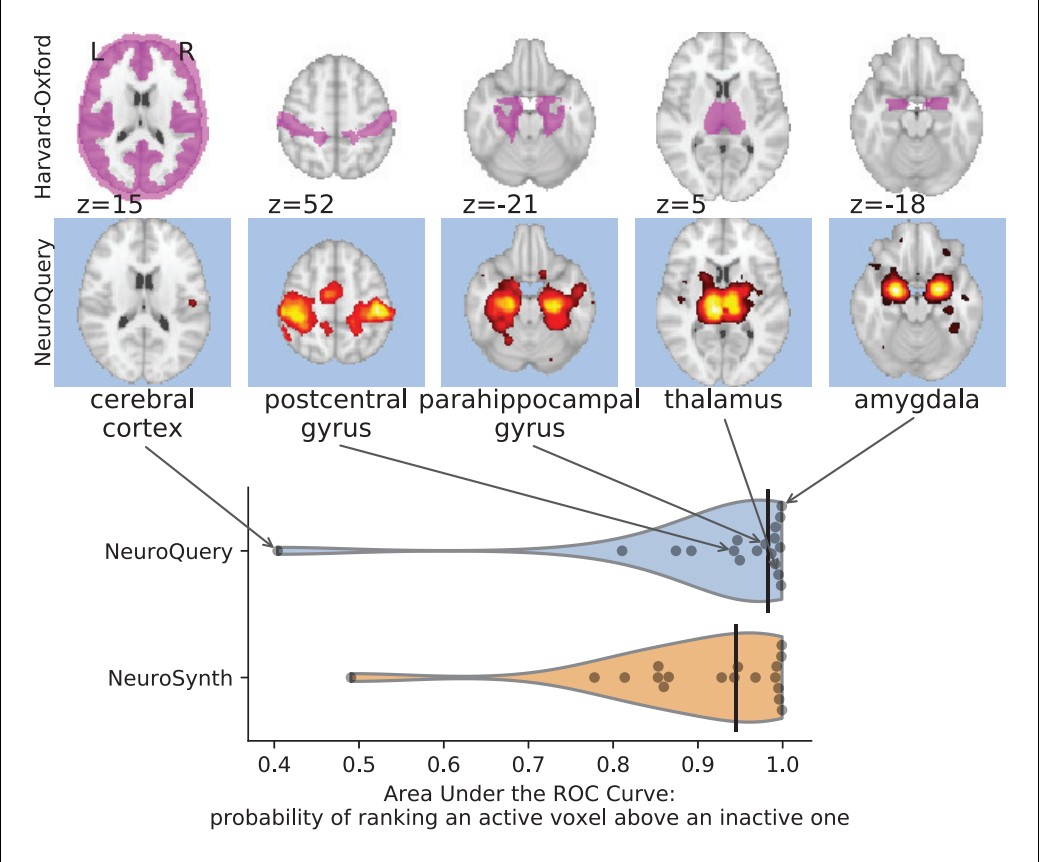

**Figure 14.** Comparison of predictions with regions of the Harvard-Oxford anatomical atlas. Labels of the Harvard-Oxford anatomical atlas present in NeuroSynth's vocabulary are fed to NeuroSynth and NeuroQuery. The meta-analytic maps are compared to the manually segmented reference by measuring the Area Under the ROC Curve. The black vertical bars show the median. Both NeuroSynth and NeuroQuery achieve a median AUC above 0.9. On the top, we show the atlas region (first row) and NeuroQuery prediction (second row) for the quartiles of the NeuroQuery AUC scores.

experts. For simplicity, atlas labels absent from NeuroSynth's vocabulary are discarded. For the remaining 18 labels, we compute the Area Under the ROC Curve of the maps produced by each meta-analytic tool. This experiment is therefore identical to the one presented in Section 'Comparison with the BrainPedia IBMA study', except that the reference ground truth is a manually segmented anatomical atlas, and that we do not consider reformulating the labels. GCLDA is not used in this experiment as the trained model distributed by the authors does not recognize anatomical terms. We observe that both NeuroSynth and NeuroQuery match closely the reference atlas, with a median AUC above 0.9, as seen in *Figure 14*.

## Comparison with NeuroSynth on terms with strong activations
As NeuroSynth performs a statistical test, when a term has a strong link with brain activity and is popular enough for NeuroSynth to detect many activations, the resulting map is trustworthy and can be used as a reference. Moreover, it is a well-established tool that has been adopted by the neuro-imaging community. Here, we verify that when a term is well captured by NeuroSynth, NeuroQuery predicts a similar brain map. To identify terms that NeuroSynth captures well, we compute the NeuroSynth maps for all the terms in NeuroSynth's vocabulary. We use the Benjamini-Hochberg procedure to threshold the maps, controlling the FDR at 1%. We then select the 200 maps with the largest number of active (above-threshold) voxels. We use these activation maps as a reference to which we compare the NeuroQuery prediction. For each term, we compute the Area Under the ROC Curve: the probability that a voxel that is active in the NeuroSynth map will have a higher value

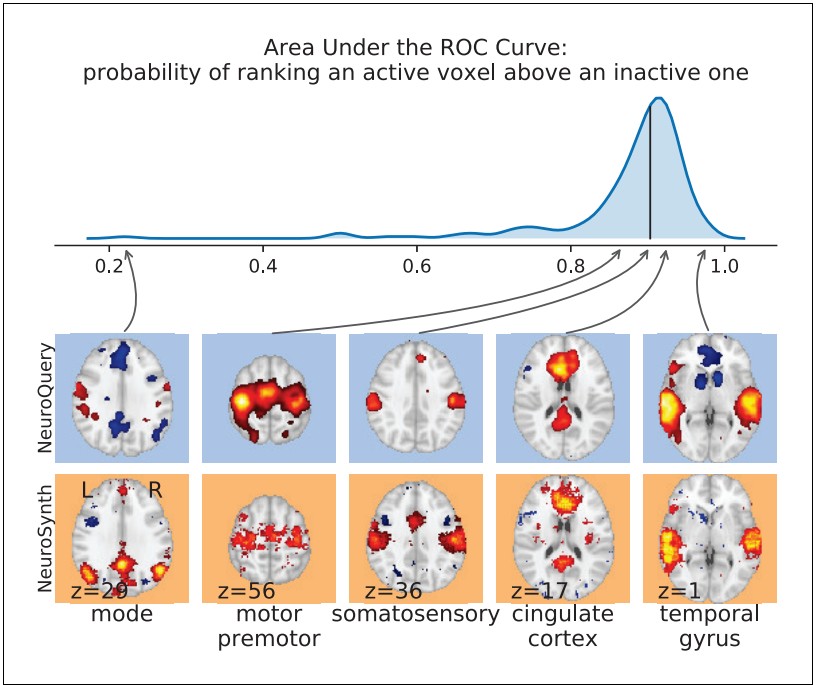

**Figure 15.** Comparison with NeuroSynth. NeuroSynth maps are thresholded controlling the FDR at 1%. The 200 words with the largest number of active voxels are selected and NeuroQuery predictions are compared to the NeuroSynth activations by computing the Area Under the ROC Curve. The distribution of the AUC is shown on the top. The vertical black line shows the median (0.90). On the bottom, we show the NeuroQuery maps (first row) and NeuroSynth activations (second row) for the quartiles of the NeuroQuery AUC scores.

in the NeuroQuery prediction than an inactive voxel. We find that NeuroQuery and NeuroSynth's maps coincide well, with a median AUC of 0.90. The distribution of the AUC and the brain map corresponding to each quartile are shown in *Figure 15*.

## The NeuroQuery publication corpus and associated vocabulary
### Word occurrence frequencies across the corpus
The challenge: most words are rare

As shown on *Figure 16* right, most words occur in very few documents, which is why correctly mapping rare words is important. The problem of rare words is more severe in the NeuroSynth corpus, which contains only the abstracts. As the NeuroQuery corpus contains the full text of the articles (around 20 times more text), more occurrences of a unique term in a document are observed, as shown in *Figure 16* left, and in *Figure 17* for a few example terms.

Document set intersections lack statistical power. For example, 'face perception' occurs in 413 articles, and 'dementia' in 1312. 1703 articles contain at least one of these words and could be used for a multivariate regression's prediction for the query 'face perception and dementia'. Indeed, denoting $c$ the dual coefficients of the ridge regression and $X$ the training design matrix, the prediction for a query $q$ is $q^t X^t c$, and any document that has a nonzero dot product with the query can participate in the prediction. However, only 22 documents contain both terms and would be used with the classical meta-analysis selection, which would lack statistical power and fail to produce meaningful results. Exact matches of multi-word expressions such as 'creative problem solving', ' facial trustworthiness recognition ', 'positive feedback processing', 'potential monetary reward', 'visual word recognition' (all cognitive atlas concepts, all occurring in less than 5/10 000 full-text articles), are very rare – and classical meta-analysis thus cannot produce results for such expressions. In *Figure 18*, we compare the frequency of multi-word expressions from our vocabulary (such as 'face recognition') with the frequency of their constituent words. Being able to combine words in an additive fashion is crucial to encode such expressions into brain space.

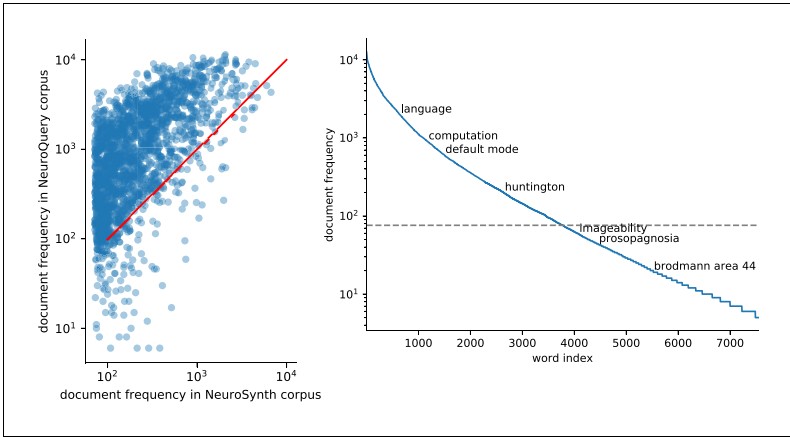

**Figure 16.** Right: benefit of using full-text articles. Document frequencies (number of documents in which a word appears) for terms from the NeuroSynth vocabulary, in the NeuroSynth corpus (*x* axis) and the NeuroQuery corpus (*y* axis). Words appear in much fewer documents in the NeuroSynth corpus because it only contains abstracts. Even when considering only terms present in the NeuroSynth vocabulary, the NeuroQuery corpus contains over 3M term-study associations – 4.6 times more than NeuroSynth. Left: Most terms occur in few documents Plot of the document frequencies in the NeuroQuery corpus, for terms in the vocabulary, sorted in decreasing order. While some terms are very frequent, occurring in over 12 000 articles, most are very rare: half occur in less than 76 (out of 14 000) articles.

## The choice of vocabulary
### Details on the Medical Subject Headings
The Medical Subject Headings (MeSH) are concerned with all of medicine. We only included in NeuroQuery's vocabulary the parts of this graph that are relevant for neuroscience and psychology. Here we list the branches of Medical Subject Headings (MeSH) that we included in our vocabulary:

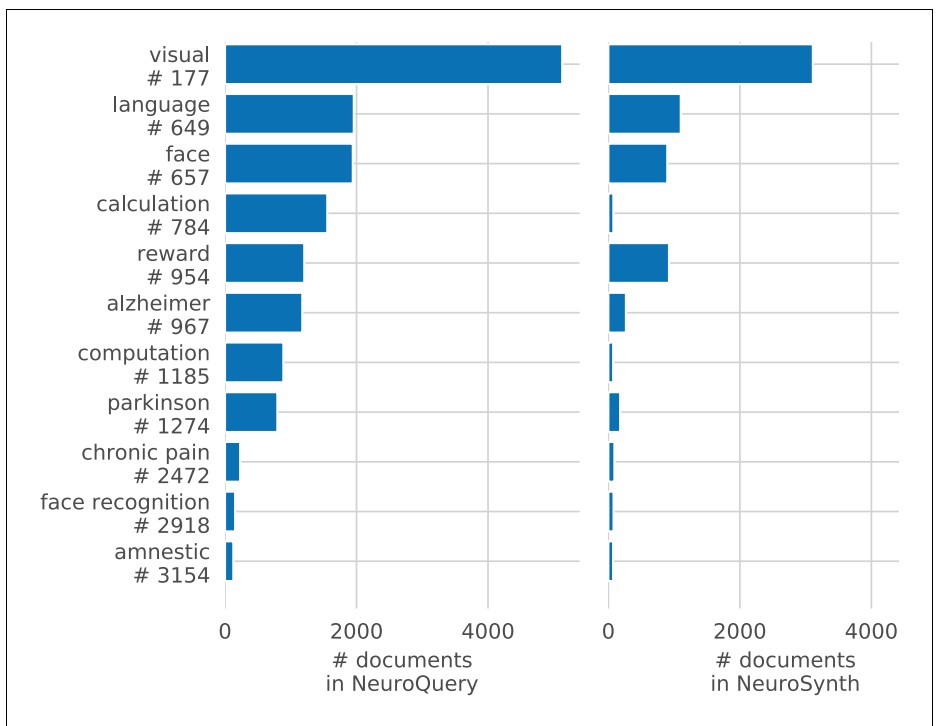

**Figure 17.** Document frequencies for some example words, in NeuroQuery's and NeuroSynth's corpora.

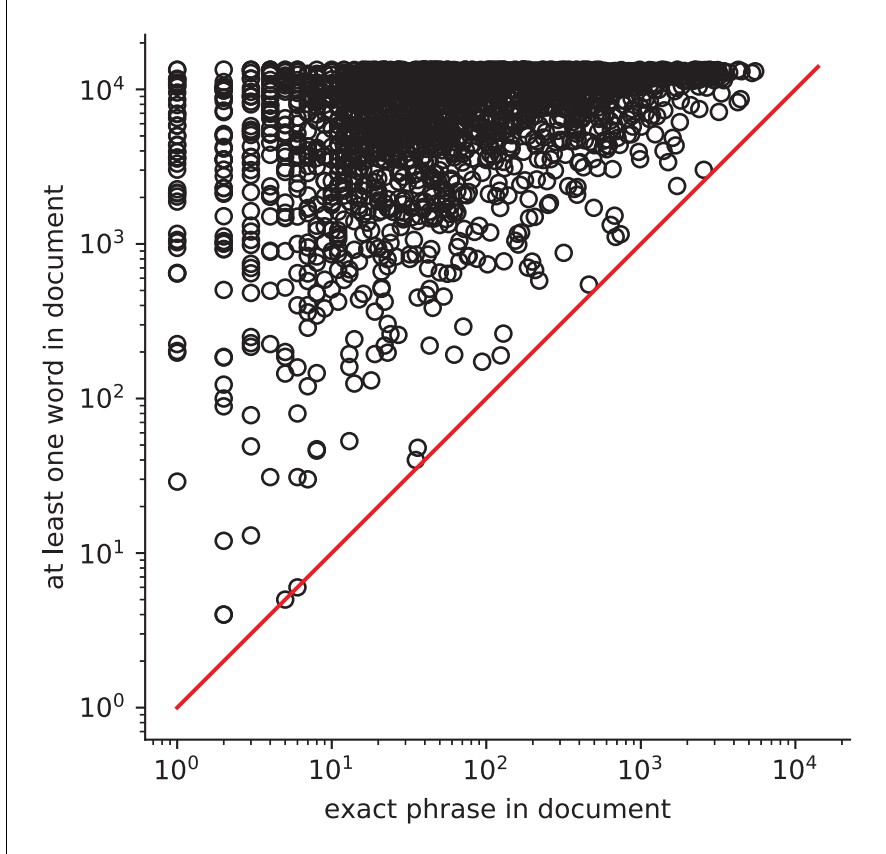

**Figure 18.** Occurrences of phrases versus its constituents How often a phrase from the vocabulary (e.g. 'face recognition') occurs, versus at least one of its constituent words (e.g. 'face'). Expressions involving several words are typically very rare.

Neuroanatomy: 'A08.186.211'
Neurological disorders: 'C10.114', 'C10.177', 'C10.228', 'C10.281', 'C10.292', 'C10.314', 'C10.500', 'C10.551', 'C10.562', 'C10.574', 'C10.597', 'C10.668', 'C10.720', 'C10.803', 'C10.886', 'C10.900'
Psychology: 'F02.463', 'F02.830', 'F03', 'F01.058', 'F01.100', 'F01.145', 'F01.318', 'F01.393', 'F01.470', 'F01.510', 'F01.525', 'F01.590', 'F01.658', 'F01.700', 'F01.752', 'F01.829', 'F01.914'

Many MeSH terms are too rare to be part of NeuroQuery's vocabulary. Some are too specific, e. g. 'Diffuse Neurofibrillary Tangles with Calcification'. More importantly, many terms are absent

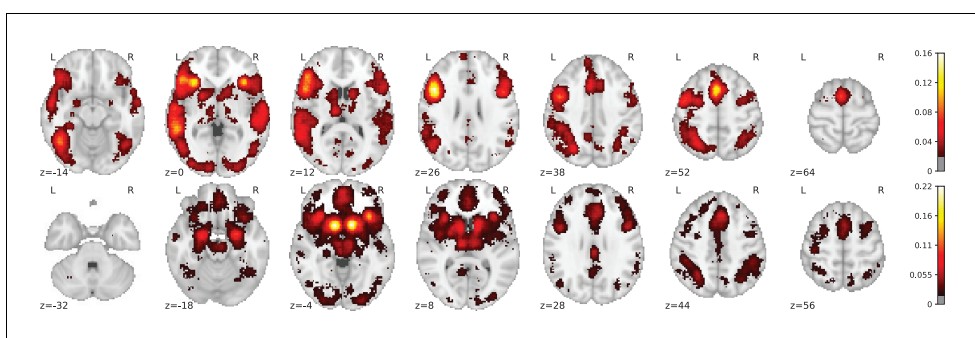

**Figure 19.** NeuroSynth posterior probability maps for 'language' (top) and 'reward' (bottom), using the full corpus.

because for each heading, MeSH provides many *Entry Terms* – various ways to refer to a concept, some of which are almost never used in practice in the text of publications. For example Neuro-Query recognizes the MeSH *Preferred Term* 'Frontotemporal Dementia' but not some of its variations (https://meshb.nlm.nih.gov/record/ui?ui=D057180) such as 'Dementia, Frontotemporal', 'Disinhibition-Dementia-Parkinsonism-Amyotrophy Complex', or 'HDDD1'. Note that even when absent from the vocabulary as single phrases, many of these variations can be parsed as a combination of several terms, resulting in a similar brain map as the one obtained for the preferred term.

## Atlas labels included in the vocabulary
The labels from the 12 atlases shown in *Table 4* were included in the NeuroQuery vocabulary.

### NeuroSynth posterior probability maps
The NeuroSynth maps shown in *Figure 6* are the NeuroSynth 'association test' maps. For completeness, in *Figure 19* we show the other kind of map that NeuroSynth can produce, called 'posterior probability' maps.

## Acknowledgements
JD acknowledges funding from Digiteo under project Metacog (2016-1270D). RP received funding from the US National Science Foundation (Award # OAC-1649658). TY acknowledges funding from NIH under grant number R01MH096906. BT received funding from the European Union's Horizon 2020 Research and Innovation Programme under Grant Agreement No. 785907 (HBP SGA2) and No 826421 (VirtualbrainCloud). FS acknowledges funding from ANR via grant ANR-16- CE23-0007-01 ('DICOS'). GV was partially funded by the Canada First Research Excellence Fund, awarded to McGill University for the Healthy Brains for Healthy Lives initiative. We also thank the reviewers, including Tor D Wager, for their suggestions that improved the manuscript.

## Additional information

### Competing interests
Gaël Varoquaux: Reviewing editor, *eLife*. The other authors declare that no competing interests exist.

### Funding

| Funder | Grant reference number | Author |
| --- | --- | --- |
| Digiteo | 2016-1270D - Projet MetaCog | Jérôme Dockès |
| National Institutes of Health | R01MH096906 | Tal Yarkoni |
| Agence Nationale de la Recherche | ANR-16- CE23-0007-01 | Fabian Suchanek |
| H2020 European Research Council | 785907 (HBP SGA2) | Bertrand Thirion |
| H2020 European Research Council | 826421 (VirtualbrainCloud) | Bertrand Thirion |
| Canada First Research Excellence Fund | Healthy Brains for Healthy Lives initiative | Gael Varoquaux |
| National Science Foundation | OAC-1649658 | Russell A Poldrack |

The funders had no role in study design, data collection and interpretation, or the decision to submit the work for publication.

## Author contributions
Jérôme Dockès, Conceptualization, Resources, Data curation, Software, Formal analysis, Validation, Investigation, Visualization, Methodology; Russell A Poldrack, Conceptualization, Supervision, Methodology; Romain Primet, Hande Gözükan, Software, Visualization; Tal Yarkoni, Conceptualization, Validation; Fabian Suchanek, Conceptualization, Supervision, Validation, Visualization, Project administration; Bertrand Thirion, Conceptualization, Supervision, Funding acquisition, Project administration; Gaël Varoquaux, Conceptualization, Software, Supervision, Funding acquisition, Validation, Visualization, Methodology, Project administration

## Author ORCIDs
Jérôme Dockès (iD) https://orcid.org/0000-0002-5304-2496
Russell A Poldrack (iD) https://orcid.org/0000-0001-6755-0259
Gaël Varoquaux (iD) https://orcid.org/0000-0003-1076-5122

## Decision letter and Author response
Decision letter https://doi.org/10.7554/eLife.53385.sa1
Author response https://doi.org/10.7554/eLife.53385.sa2

---

# Additional files

## Supplementary files
• Source data 1. Source code for figures and tables.
• Transparent reporting form

## Data availability
All the data that we can share without violating copyright (including word counts of publications) have been shared on https://github.com/neuroquery/ (copy archived at https://github.com/elifesciences-publications/neuroquery) alongside with the analysis scripts. Everything is readily downloadable without any authorization or login required. For each figure and table, the data directly used to generate it is made available in a separate zip file.

The following dataset was generated:

| Author(s) | Year | Dataset title | Dataset URL | Database and Identifier |
| --- | --- | --- | --- | --- |
| Dockès J, Varoquaux G | 2019 | NeuroQuery | https://github.com/neuroquery/neuroquery_data | GitHub, neuroquery_data |

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
