## [Decision Letter]

**Acceptance summary:**

This paper describes NeuroQuery, a new approach to automated meta-analysis of the neuroimaging literature. It is demonstrably superior for many purposes, particularly as a starting point for constraining predictive models and as regions or patterns of interest in new studies. We believe that this will be a very widely used tool. It's also a tremendous amount of work to build and validate, and very few groups have both the skills and motivation to build this and make it accessible to the broad neuroscience/neuroimaging community.

**Decision letter after peer review:**

Thank you for submitting your article "NeuroQuery: comprehensive meta-analysis of human brain mapping" for consideration by *eLife*. Your article has been reviewed by three peer reviewers, and the evaluation has been overseen by a Reviewing Editor and Christian Büchel as the Senior Editor. The following individuals involved in review of your submission have agreed to reveal their identity: Tor D. Wager (Reviewer #1).

The reviewers have discussed the reviews with one another and the Reviewing Editor has drafted this decision to help you prepare a revised submission.

Summary:

This paper describes NeuroQuery, a new approach to automated meta-analysis of the neuroimaging literature. It is demonstrably superior for many purposes – particularly as a starting point for constraining predictive models and as regions or patterns of interest in new studies. We believe that this will be a very widely used, and cited, tool. It's also a tremendous amount of work to build and validate, and very few groups have both the skills and motivation to build this and make it accessible to the broad neuroscience/neuroimaging community.

There are a number of exemplary features of this work, including:

– A full implementation that is openly shared on Github, including source code, models, and data

– A fully functional web interface that runs simply and beautifully

– Several types of validation of the model's performance, including (a) stability with rare terms listed in few studies, (b) reproducibility of maps with limited data, (c) better encoding/prediction of brain maps associated with terms.

This work dramatically expands the vocabulary of search terms that can be used in neuroimaging meta-analyses. We believe this is going to be really useful. The model framework is also interesting and motivated by some careful considerations in terms of how the model should be affected by certain classes of rare terms, features included in the model, etc. The bottom line for users is a better set of predictive maps. While there are many potential varieties of such models and one could second-guess some particular choices, the beauty of the authors' approach is that the code is available for others to try out variations and build/validate a different model based on their sensibilities and preferences.

Essential revisions:

1) We believe that the validations are insufficient for some of the claims. The authors should either expand their experiments to validate these claims or tone down their claims.

A) "NeuroQuery, can assemble results from the literature into a brain map based on an arbitrary query." This statement can be mis-interpreted to mean that users can enter any terms they want. My understanding is that the query has to comprise words from the 7547 dictionary words. Words outside the dictionary are ignored.

B) Introduction: "For example, some rare diseases, or a task involving a combination of mental processes that have been studied separately, but never – or rarely – together." This suggests NeuroQuery can do this with precision, but the authors have not experimentally demonstrated that activation maps involving "combination of mental processes that have been studied separately, but never together" can be accurately predicted. To validate this, the authors should consider cases involving compound mental processes (e.g., "auditory working memory") and remove all experiments involving both "auditory" + "working memory". Care has to be taken to also remove experiments using words similar to "auditory" + "working memory", so that experiments such as "auditory n-back" are also removed, even though the experiments did not explicitly use the term "working memory". Note that the entire experiments should be removed, rather than just the specific words. The authors can then re-train their model and show whether the query "auditory working memory" yields activation similar to the removed experiments involving auditory working memory.

C) For subsection “NeuroQuery can map rare or difficult concepts”, it's important to differentiate between a concept that is rarely studied versus a rare word that seldom appears in the literature even though variations of the word might be widely studied. The whole section seems to imply that the NeuroQuery is able to accomplish the former very well, but in the experiment, they chose frequent terms (e.g., language) and progressively delete them from their dataset, thus they are really testing the latter. To really test the former, rather than just deleting the word "language" from the documents, they should delete entire documents containing "language" and/or other terms similar to language (e.g., semantic).

2) Figure 8 (right panel) should be in the main text in addition to (or replacing) Figure 6. While the log likelihood measure is valid, the "absolute" measure is much more helpful to the users of knowing how much to trust the results. Along this note, it is somewhat concerning that the overall accuracy is only 70% – how much should a user trust a tool with a 70% accuracy? However, this is perhaps underselling NeuroQuery because coordinates tend to be sparse, so just based on the reported coordinates, this classification task might simply be very hard. What might be more useful would be for the user to get a sense of how much the NeuroQuery maps actually matches real activation maps. Can the authors perform the same experiment, but using real activation maps from NeuroVault or their own Individual Brain Charting dataset? This is just a suggestion. The authors can choose to just discuss this issue.

3) Results section: some pieces of argumentation presented here are not fully convincing. The authors propose that: "Term co-occurrences, on the other hand, are more consistent and reliable, and capture semantic relationships [Turney and Pantel, 2010]". Most of the brain mapping literature is made of attempts to differentiate cognitive processes inferred from the study of human behaviour such as "proactive control VS retroactive control", "recollection VS familiarity", "face perception VS place perception", "positive emotion VS negative emotion". In that context, it is unclear how terms co-occurrence for example between "face" and "place" would be more consistent and reliable than "face" alone. Term co-occurrence mainly reflects what type of concepts tend to be studied together.

4) The authors suggest that "It would require hundreds of studies to detect a peak activation pattern for "aphasia". It is not clear what do the authors mean here. Aphasia is a clinical construct referring to a behavioural disturbance. It is defined as "an inability (or impaired ability) to produce or understand speech". Accordingly, we don't see how one could search for a peak of activation related to aphasia? The line of argumentation could be clarified here by starting with concrete examples. Perhaps an example the authors may refer to is that, starting from a clinical point of view, researchers may want to investigate brain activation related to the processes of language production in order to better understand dysfunctions such those seen in aphasia. Accordingly, the related publication will probably mention "aphasia" "language production" "phonological output lexicon" etc… and this pattern of terms could in turn be linked to similar language related terms in other publications. An alternative guess is that maybe the authors actually refer here to lesion studies that have, for example, searched for correlation between grey matter volume atrophy and aphasia?

5) We have some suggestions for improving the Abstract/Introduction

A) Explaining how a predictive framework allows maps to be generated by generalizing from terms that are well studied ("faces") to those that are less well studied and inaccessible to traditional meta-analyses ("prosopagnosia"). And explaining that this is good for some use cases (generating sensible ROIs/making guesses about where future studies of prosopagnosia might activate), but not good for others (e.g., knowing that a particular area is (de)activated in studies of prosopagnosia).

B) Can you provide a bit more context on previous topic models (e.g., Nielsen/Hansen NNMF "bag of words" work from way back, Poldrack/Yarkoni topic models, GCLDA) and how this approach is different. After reading the Introduction, readers have a sense of what Neuroquery does (its goal), but not how it actually does it (no algorithmic insight). For example, Neuroquery "infers semantic similarities across terms used in the literature", but so does a topic model (which is not mentioned/cited). We recommend including more description early on of what the algorithmic differences are relative to previous work that confer advantages. This is explained later (with some redundancy across sections), but more up front would be helpful.

C) In the Introduction, the authors argue that meta-analyses are limited primarily by performance of in-sample statistical inference (they "model all studies asking the same question") and lack of use of a predictive framework. While there is great value in the current work, we don't think that is strictly true. Other meta-analyses have taken a predictive approach, in a limited way, and also modeled differences across study categories. The Naive Bayes classifier analyses in the Neurosynth paper (Yarkoni, 2011) are an example. And Neurosynth considers topics across tasks and fields of study. Analyzing studies by single search terms is an important limitation (topic models by Poldrack and Yarkoni help; perhaps acknowledge their 2014 paper). Viewing the paper as a whole, we understand why the authors focus on prediction vs. inference in the Introduction, but it's hard to understand how this other work fits in without having gone through the whole paper.

D) Likewise, the emphasis on the ability to interpolate struck us as odd. We understand this as generalizing based on semantic smoothing. This is useful, but we don't think what you meant was really clearly explained in the Introduction.

E) The authors make a compelling case that current meta-analyses are limited. The Pinho 2018 example is very helpful: An automated meta-analysis of all the studies performing the same or a similar contrast is not currently possible, primarily because we have lacked the tools to parse the natural language text in articles well enough to identify an "equivalence class" of functional contrasts.

F) "For example, some rare diseases, or a task involving a combination of mental processes that have been studied separately, but never – or rarely – together." Could the authors provide a concrete example?

6) We have some suggestions for improving the clarity of the Results section:

A) “NeuroQuery relies on a corpus of 13459 full-text neuroimaging publications”

What type of neuroimaging publications? task-based activations experiments? If so, of which type: PET and fMRI or only fMRI-based? does it also include structural MRI studies? does it include all types of populations (healthy adults, elderly, patients studies)?

B) It is not clear how to read Figure 1. What do the length and width of the color and grey lines represent? could this figure be more elaborated to integrate additional aspects of the procedure? e.g. each of the three brain slice has a color that seems to reflect their respective association with the term, how is all that information combine to a single brain pattern?

C) Terminological precision: Results paragraph one: the term "area" is usually reserved for brain territories defined based on microstructure features (e.g. Area 4). Here the authors refer to a spatial location in the brain (or maybe a zone of homogeneity with regards to a specific feature), so they should prefer the term "brain region" for that purpose.

D) Finally, the supplement material provides several quantitative evidence of significant improvement compared to Neurosynth, such as:

– "our method extracted false coordinates from fewer articles: 3 / 40 articles have at least one false location in our dataset, against 20 for NeuroSynth"

– "In terms of raw amount of text, this corpus is 20 times larger than NeuroSynth's"

– "Compared with NeuroSynth, NeuroQuery's extraction method reduced the number of articles with incorrect coordinates (false positives) by a factor of 7, and the number of articles with missing coordinates (false negatives) by a factor of 3 (Table 2)."

All these aspects could be summarized together with the more qualitative aspects in a table to emphasize the significant improvements over Neurosynth.

7) In the Discussion, it would be very helpful to give readers some intuition for when NeuroQuery will not yield sensible results and when/how exactly it should be used (e.g., even a table of use cases that would be appropriate and inappropriate) – and how to interpret carefully (e.g., look at the semantic loadings, and if there is one anatomical term that dominates, realize that you're essentially getting a map for that brain region). The ADHD example is useful but doesn't really cover the space of principles/use cases. Here are some possible examples we have thought of:

A) Some particular limitations may arise from the predictive nature of neuroquery, which may be less intuitive to many readers. For example, if I put in "aphasia", I will get map for "language", because aphasia is semantically close to language. This is very sensible, but users should not, of course, take this as a map of "aphasia" to be related other terms and used in inference. Users might "discover" that aphasia patterns are very closely related to "language" patterns and make an inference about co-localization of healthy and abnormal function. Of course, it's not your responsibility to control all kinds of potential misuse. But pointers would be helpful to avoid another, e.g., "#cingulategate" (Lieberman et al., 2016 PNAS).

B) For example, let's consider again the case about "combination of mental processes that have been studied separately, but never together". From my understanding of the algorithm, suppose users query "auditory" + "working memory", basically the prediction will be a linear combination of activation maps from "auditory" and "working memory" (+ similar terms due to the smoothing/query expansion). As such, this assumes that compound mental processes yield activations that are linear combination of activation maps of individual mental processes. This should be made clear.

C) Playing around with NeuroQuery, there are some queries that generate obviously wrong results. For example, "autobiographical memory" should probably yield the default network, but we getting hippocampal/retrosplenial activation instead. This presumably happens because NeuroQuery "expanded" the query to become memory because "autobiographical memory" is not one of the 200 keywords? Interesting that NeuroSynth does get it correct (https://neurosynth.org/analyses/terms/autobiographical%20memory/).

D) Perhaps a brief discussion of other limitations would be helpful. We submit that some of the fundamental problems are those not easily solved – that we usually perform meta-analyses based on studies of the same nominal task type (e.g., N-back), and sometimes minor variations in task structure can yield divergent findings. We don't know what all the dimensions are yet. This problem goes far beyond the challenge of establishing a set of consensus labels for task types and relevant cognitive processes. In short, we don't really even know what task features to label yet in many cases, and they don't combine additively. A stop-signal task with one adaptive random walk may be different than one with four, as it allows a different type of cognitive strategy.

E) When to use it: For common terms, meta-analysis (e.g., Neurosynth) does very well (e.g., Figure 4). When would the authors recommend using NeuroQuery over another meta-analytic tool? Maybe they could provide a summary of use cases and conditions (e.g., when few studies of a term/topic are available). Also see point (C).

8) Discussion of other approaches: We submit that the field has become tracked into a relatively narrow space of the possible options and techniques for meta-analysis, based on local analysis of coordinates in MKDA/ALE. Alternatives could be mentioned as potential future directions. For example, early work explored clustering of spatial locations and spatial discriminant tests (e.g., Wager et al., 2002, 2004, 2005), and later work has explored spatial models (e.g., Kang, 2011, Kang, 2014, Wager, 2015) and more advanced co-activation models (Xue, 2014). While this is obviously beyond the scope of the present paper, future work might consider models of spatial co-activation when generating predictions and inferences about meta-analytic maps.

9) The methods are a bit unclear:

A) Significant details about methodologies are missing. How are term frequency and inverse document frequency computed?

B) Equation 5: How is σ^ij computed? Square root of the entries of equation 8?

C) Equation 5: What is the difference between σ^_i_ and σ^_ij_? σ^_i_ is a column/row of σ^_ij_?

D) Y_j_ is the j-th column not i-th column of Y?

E) M is a v x n matrix, so equation 8 is a v x v matrix? We are confused how this maps to σ^_ij_.

F) In Equation 10, is ||U||_1 just the sum of absolute values of all entries in U?

G) What is k set to be in equation 10?

H) "More than 72% of the time, NeuroQuery's output has a higher Pearson correlation with the correct map than with the negative example" – "correct map" refers to the KDE density maps?

10) Subsection “Smoothing: regularization at test time” is hard to read. It would be helpful if the authors explain the intuition behind the different steps and what the different matrices represent. For example, it might be helpful to explain that each row of V can be thought of as the association of words with topic k, so a higher value for row i, column j suggests the j-th dictionary word is more strongly related to the topic k. As another example, the authors should also unpack subsection “Smoothing: regularization at test time”: why do we take V and scale with 𝑛_𝑖,𝑖_ and then compute C and then l1-normalizing the rows of C to produce T and then finally S. What does each step try to do? We guess roughly speaking VV^T is like a co-occurrence matrix (how likely are two words likely to appear together?), but we are not sure why we have to do the extra normalization with 𝑛_𝑖,𝑖_, l1-normalization, etc.

---

## [Author Response]

Essential revisions:1) We believe that the validations are insufficient for some of the claims. The authors should either expand their experiments to validate these claims or tone down their claims.

We have both added evidence, and toned down claims where they were not justified.

A) "NeuroQuery, can assemble results from the literature into a brain map based on an arbitrary query." This statement can be mis-interpreted to mean that users can enter any terms they want. My understanding is that the query has to comprise words from the 7547 dictionary words. Words outside the dictionary are ignored.

Indeed, we made this statement more precise.

B) Introduction: "For example, some rare diseases, or a task involving a combination of mental processes that have been studied separately, but never – or rarely – together." This suggests NeuroQuery can do this with precision, but the authors have not experimentally demonstrated that activation maps involving "combination of mental processes that have been studied separately, but never together" can be accurately predicted. To validate this, the authors should consider cases involving compound mental processes (e.g., "auditory working memory") and remove all experiments involving both "auditory" + "working memory". Care has to be taken to also remove experiments using words similar to "auditory" + "working memory", so that experiments such as "auditory n-back" are also removed, even though the experiments did not explicitly use the term "working memory". Note that the entire experiments should be removed, rather than just the specific words. The authors can then re-train their model and show whether the query "auditory working memory" yields activation similar to the removed experiments involving auditory working memory.

We thank the reviewers for outlining this aspect of the contribution that was not sufficiently validated. We have added a section (“NeuroQuery can map new combinations of concepts”) with new experiments to showcase that NeuroQuery indeed can map unseen combinations of terms, following the reviewers’ suggestion to fit NeuroQuery on a corpus of study excluding studies of the combination of two given terms. We have first shown the maps for qualitative assessment on the combination of "color" and "distance" (Figure 3). We have also performed a quantitative assessment leaving random pairs of terms unseen (also section “NeuroQuery can map new combinations of concepts”). When choosing the random pairs of terms, we did not remove related words as it is very difficult to do in an automated way, in particular it runs the risk of depleting the corpus by removing too many studies. However, for the specific case of "color" and "distance", we do not see any clear synonyms or alternate way of referring to the same exact mental process in our vocabulary. Hence we feel that the corresponding figure addresses the reviewers’ suggestion.

C) For subsection “NeuroQuery can map rare or difficult concepts”, it's important to differentiate between a concept that is rarely studied versus a rare word that seldom appears in the literature even though variations of the word might be widely studied. The whole section seems to imply that the NeuroQuery is able to accomplish the former very well, but in the experiment, they chose frequent terms (e.g., language) and progressively delete them from their dataset, thus they are really testing the latter. To really test the former, rather than just deleting the word "language" from the documents, they should delete entire documents containing "language" and/or other terms similar to language (e.g., semantic).

We have added at the end of subsection “NeuroQuery can map rare or difficult concepts” a paragraph making explicit the difference between these two aspects. We explain how NeuroQuery improves also for seldom-studied concepts. However, we can only present face validity as evidence for this claim, for lack of solid ground truth on these terms. We have added maps for "color" and "Huntington" on Figure 4: these terms are rare (the number of occurrences is shown on Figure 4), and they do not have synonyms in our vocabulary and therefore the quality of the maps that they produce is evidence that NeuroQuery maps rare concepts.

2) Figure 8 (right panel) should be in the main text in addition to (or replacing) Figure 6. While the log likelihood measure is valid, the "absolute" measure is much more helpful to the users of knowing how much to trust the results. Along this note, it is somewhat concerning that the overall accuracy is only 70% – how much should a user trust a tool with a 70% accuracy? However, this is perhaps underselling NeuroQuery because coordinates tend to be sparse, so just based on the reported coordinates, this classification task might simply be very hard. What might be more useful would be for the user to get a sense of how much the NeuroQuery maps actually matches real activation maps. Can the authors perform the same experiment, but using real activation maps from NeuroVault or their own Individual Brain Charting dataset? This is just a suggestion. The authors can choose to just discuss this issue.

We have moved the "mix and match" results to the main manuscript. While the reviewer finds that 72% accuracy in distinguishing studies is a disappointing result, we fear that it might not be a limitation of the statistical modeling, but rather a measure of noise in reported results. Indeed, the amount of variability in reported results can be gauged visually with the Neuroquery only interface: for a given query, the coordinates reported in the most relevant publications can be seen by hovering over the list, below the predicted map. Such an experiment shows a great variability in publications on similar topics.

We thank the reviewers for suggesting to compare NeuroQuery predictions to real activation maps. We have addressed it in subsection “NeuroQuery maps reflect well other meta-analytic maps”. We considered using the IBC dataset as suggested. IBC activation maps do not suffer from the sparsity of peak coordinates. However, as many individual fMRI maps, they are also tainted by an important amount of noise.

To confront NeuroQuery to a more reliable ground truth, we therefore preferred to rely on an Image-Based Meta-Analysis study. We compared NeuroQuery predictions to maps of the BrainPedia study, published last year by some authors of the current submission. This dataset is interesting because it covers a wide variety of cognitive concepts, the studies included in the meta-analysis were chosen manually, and the meta-analytic maps were carefully labelled. The maps and labels from this publication were uploaded to NeuroVault in December, 2018.

When reformulating the BrainPedia labels to match NeuroSynth’s vocabulary, both NeuroSynth and NeuroQuery matched well the IBMA results, with a median AUC of respectively 0.8 and 0.9. Importantly, while reformulating the labels is necessary to obtain results from NeuroSynth, NeuroQuery coped well with the original, uncurated labels, obtaining a median AUC of 0.8. This illustrates NeuroQuery’s capacity to handle less restricted queries, that enables it to be integrated in more automated pipelines.

Although they come from an IBMA of manually selected studies, BrainPedia maps still contain some noise. To perform a sanity check relying on an excellent ground truth, we obtained maps from NeuroSynth and NeuroQuery for labels of the Harvard-Oxford structural atlases. The CBMA maps matched well the regions manually segmented by experts, with a median AUC above 0.9.

Finally, we compared NeuroQuery predictions to NeuroSynth activations for terms that NeuroSynth captures well (the 200 words that produce the largest number of activations with NeuroSynth). We found that the results of NeuroQuery and NeuroSynth for these terms were close, with a median AUC of 0.9. Results from these experiments are presented in subsection “NeuroQuery maps reflect well other meta-analytic maps”.

3) Results section: some pieces of argumentation presented here are not fully convincing. The authors propose that: "Term co-occurrences, on the other hand, are more consistent and reliable, and capture semantic relationships [Turney and Pantel, 2010]". Most of the brain mapping literature is made of attempts to differentiate cognitive processes inferred from the study of human behaviour such as "proactive control VS retroactive control", "recollection VS familiarity", "face perception VS place perception", "positive emotion VS negative emotion". In that context, it is unclear how terms co-occurrence for example between "face" and "place" would be more consistent and reliable than "face" alone. Term co-occurrence mainly reflects what type of concepts tend to be studied together.

An important aspect of the semantic techniques that we use is that they give a continuous measure of relatedness. For this reason, we think that the use of oppositions in neuroimaging is not a fundamental roadblock: terms that are often studied in opposition are different, but they are related, on the bigger picture of cognition. All the examples listed by the reviewer are pairs drawn from the same subfield of cognition, which would be encoded as a specific branch in a cognitive ontology. The co-occurrences thus help meta-analysis: "recollection" is closer to "familiarity" than to "positive emotion". Yet, research in distributional semantics (which lead for instance to the famous "word2vec" model) has shown that they will have a less strong co-occurrence than exact synonyms, such as "acc" and "anterior cingulate cortex". We have added these considerations in the middle of subsection “Overview of the NeuroQuery model”: "The strength of these relationships encode semantic proximity, from very strong for synonyms that occur in statistically identical contexts, to weaker for different yet related mental processes that are often studied one opposed to the other."

4) The authors suggest that "It would require hundreds of studies to detect a peak activation pattern for "aphasia". It is not clear what do the authors mean here. Aphasia is a clinical construct referring to a behavioural disturbance. It is defined as "an inability (or impaired ability) to produce or understand speech". Accordingly, we don't see how one could search for a peak of activation related to aphasia? The line of argumentation could be clarified here by starting with concrete examples. Perhaps an example the authors may refer to is that, starting from a clinical point of view, researchers may want to investigate brain activation related to the processes of language production in order to better understand dysfunctions such those seen in aphasia. Accordingly, the related publication will probably mention "aphasia" "language production" "phonological output lexicon" etc… and this pattern of terms could in turn be linked to similar language related terms in other publications. An alternative guess is that maybe the authors actually refer here to lesion studies that have, for example, searched for correlation between grey matter volume atrophy and aphasia?

Indeed, the formulation here was not suitable, as "aphasia" is not a mental process but a pathology, and thus not mapped by brain activations. We changed the sentence to "It would require hundreds of studies to detect a pattern in localizations reported for’ aphasia’, as one would appear in lesion studies. But with the text of a few publications we notice that it often appears close to’ language’, which is indeed a related mental process". We choose to do a light rewording rather to discuss in depth how pathologies are captured in the brain-imaging literature: via lesions, but also by studying related mental processes, or simulated with transcranial magnetic stimulation. Indeed, we feel that this is not central to the topic of the manuscript.

5) We have some suggestions for improving the Abstract/IntroductionA) Explaining how a predictive framework allows maps to be generated by generalizing from terms that are well studied ("faces") to those that are less well studied and inaccessible to traditional meta-analyses ("prosopagnosia"). And explaining that this is good for some use cases (generating sensible ROIs/making guesses about where future studies of prosopagnosia might activate), but not good for others (e.g., knowing that a particular area is (de)activated in studies of prosopagnosia).

Indeed, these are good suggestions, that we added to the Introduction.

B) Can you provide a bit more context on previous topic models (e.g., Nielsen/Hansen NNMF "bag of words" work from way back, Poldrack/Yarkoni topic models, GCLDA) and how this approach is different. After reading the Introduction, readers have a sense of what Neuroquery does (its goal), but not how it actually does it (no algorithmic insight). For example, Neuroquery "infers semantic similarities across terms used in the literature", but so does a topic model (which is not mentioned/cited). We recommend including more description early on of what the algorithmic differences are relative to previous work that confer advantages. This is explained later (with some redundancy across sections), but more up front would be helpful.

The goal of a topic model is not to infer semantic similarity but to extract latent factors. This is a subtle difference, as much of the techniques are the same (though recent work in distributional semantics departs more and more from topic modeling), however this is the reason why we did not mention topic modeling. We have added a section on related work in the Discussion, that stresses the links and the differences to prior art. We would like our manuscript to be appealing to users of NeuroQuery, and not only to experts of meta-analysis or text mining, hence we choose to keep technicalities outside of the Introduction. Nevertheless, we added in the Introduction the mention of supervised learning to give the important keywords to the expert reader.

C) In the Introduction, the authors argue that meta-analyses are limited primarily by performance of in-sample statistical inference (they "model all studies asking the same question") and lack of use of a predictive framework. While there is great value in the current work, we don't think that is strictly true. Other meta-analyses have taken a predictive approach, in a limited way, and also modeled differences across study categories. The Naive Bayes classifier analyses in the Neurosynth paper (Yarkoni, 2011) are an example. And Neurosynth considers topics across tasks and fields of study. Analyzing studies by single search terms is an important limitation (topic models by Poldrack and Yarkoni help; perhaps acknowledge their 2014 paper). Viewing the paper as a whole, we understand why the authors focus on prediction vs. inference in the Introduction, but it's hard to understand how this other work fits in without having gone through the whole paper.

To discuss better the links with prior art, we have added a section: “Related work” Indeed, the reviewer is right that ingredients of our approach have been used for meta-analysis, and yet not in the way our work combines them. The predictive model of Neurosynth is a decoding model: it predicts (mutually exclusive) terms from brain activations, while we are doing the converse. As a result, it cannot combine information across the terms, which is the crucial aspect of our predictive framework. The Neurosynth maps are not based on the prediction of a model, but the ingredients of a model. Topic models have also been used (we believe the reviewer refers to the 2012 Poldrack paper), but again the prior art uses the model parameters to derive conclusions. Using predictions enables extrapolations, as demonstrated in our manuscript. In addition, the quality of these predictions can be directly assessed, as in the experiments that we perform.

D) Likewise, the emphasis on the ability to interpolate struck us as odd. We understand this as generalizing based on semantic smoothing. This is useful, but we don't think what you meant was really clearly explained in the Introduction.

We have reworked the part of the Introduction that mentioned interpolation to be more clearly related to the aspects that are important for our work: "They cannot model nuances across studies because they rely on in-sample statistical inference and are not designed to interpolate between studies that address related but different questions, or make predictions for unseen combination of mental processes."

E) The authors make a compelling case that current meta-analyses are limited. The Pinho 2018 example is very helpful: An automated meta-analysis of all the studies performing the same or a similar contrast is not currently possible, primarily because we have lacked the tools to parse the natural language text in articles well enough to identify an "equivalence class" of functional contrasts.

We thank the reviewers for this elegant phrasing. We have taken the liberty to add an adapted version in the Introduction, at the beginning of the paragraph on the Pinho, 2018 example.

F) "For example, some rare diseases, or a task involving a combination of mental processes that have been studied separately, but never – or rarely – together." Could the authors provide a concrete example?

We have added the specific case of "huntingon" and "agnosia": "For instance, there is evidence of agnosia in Huntington’s disease [sitek2014], but no neuroimaging study". Indeed, no publications in the dataset contain both terms, ie, no existing imaging study, but it’s not meaningless as a subject of interest (see https://www. ncbi.nlm.nih.gov/pubmed/25062855).

6) We have some suggestions for improving the clarity of the Results section:A) “NeuroQuery relies on a corpus of 13459 full-text neuroimaging publications”What type of neuroimaging publications? task-based activations experiments? If so, of which type: PET and fMRI or only fMRI-based? does it also include structural MRI studies? does it include all types of populations (healthy adults, elderly, patients studies)?

We added a short paragraph to clarify this point: "The articles are selected by querying the ESearch Entrez utility either for specific neuroimaging journals or with query strings suchs as’ fMRI’. The resulting studies are mostly based on fMRI experiments, but the dataset also contains PET or structural MRI studies. It contains studies about diverse types of populations: healthy adults, patients, elderly, children."

B) It is not clear how to read Figure 1. What do the length and width of the color and grey lines represent? could this figure be more elaborated to integrate additional aspects of the procedure? e.g. each of the three brain slice has a color that seems to reflect their respective association with the term, how is all that information combine to a single brain pattern?

We have clarified and expanded the caption of Figure 1, after "Details:.…". As is now explained in the caption, the length of the lines in the first graph shows the semantic similarity. The length of the lines in the second graph shows the weight of each term in the final prediction (the coefficient of its normalized brain map in the linear combination that produces the final prediction). The width is the same for all bars within each graph and carries no special meaning. The brain maps associated with each colored term are combined linearly, with the weights shown on the second graph, to form the final prediction shown on the far right. As this figure is meant to serve as a high-level, schematic summary of the main steps followed by NeuroQuery, we prefer not to complicate it further by integrating additional aspects of the procedure. Indeed, the exact details of the procedure are described at length in subsection “Overview of the NeuroQuery model”.

C) Terminological precision: Results paragraph one: the term "area" is usually reserved for brain territories defined based on microstructure features (e.g. Area 4). Here the authors refer to a spatial location in the brain (or maybe a zone of homogeneity with regards to a specific feature), so they should prefer the term "brain region" for that purpose.

We thank the reviewers for this correction and replaced "area" with "region" in two phrases: "which brain regions are most likely to be observed in a study."; "NeuroQuery is a statistical model that identifies brain regions".

D) Finally, the supplement material provides several quantitative evidence of significant improvement compared to Neurosynth, such as:– "our method extracted false coordinates from fewer articles: 3 / 40 articles have at least one false location in our dataset, against 20 for NeuroSynth"– "In terms of raw amount of text, this corpus is 20 times larger than NeuroSynth's"– "Compared with NeuroSynth, NeuroQuery's extraction517 method reduced the number of articles with incorrect coordinates (false positives) by a factor of 7, and the number of articles with missing coordinates (false negatives) by a factor of 3 (Table 2)."All these aspects could be summarized together with the more qualitative aspects in a table to emphasize the significant improvements over Neurosynth.

We have added Table 3 and referenced it in subsection “NeuroQuery maps reflect well other meta-analytic maps” on NeuroQuery as an openly available resource.

7) In the Discussion, it would be very helpful to give readers some intuition for when NeuroQuery will not yield sensible results and when/how exactly it should be used (e.g., even a table of use cases that would be appropriate and inappropriate) – and how to interpret carefully (e.g., look at the semantic loadings, and if there is one anatomical term that dominates, realize that you're essentially getting a map for that brain region). The ADHD example is useful but doesn't really cover the space of principles/use cases. Here are some possible examples we have thought of:A) Some particular limitations may arise from the predictive nature of neuroquery, which may be less intuitive to many readers. For example, if I put in "aphasia", I will get map for "language", because aphasia is semantically close to language. This is very sensible, but users should not, of course, take this as a map of "aphasia" to be related other terms and used in inference. Users might "discover" that aphasia patterns are very closely related to "language" patterns and make an inference about co-localization of healthy and abnormal function. Of course, it's not your responsibility to control all kinds of potential misuse. But pointers would be helpful to avoid another, e.g., "#cingulategate" (Lieberman et al., 2016 PNAS).

We have added, in the limitation paragraph, a discussion of the possible failures of NeuroQuery, as well as indications on how these failures can be detected from the user interface.

We have added a lengthy paragraph in the Discussion (beginning of subsection “Usage recommendations and limitations”) on limitations, going into the various reasons why a brain map predicted by NeuroQuery may not match the expected map. This paragraph basically explains how to craft queries to break the tool. We do hope that, given this power, the reviewers will still appreciate that the average prediction for typical query is good.

With regards to the bigger picture mentioned by the reviewers, we feel misuse of meta-analysis, as in the cingulategate, arises from a lack of understanding of the limitations of meta-analysis. Hence, we added a broad picture section (Conclusion), discussing these limitations, and positioning explicitly NeuroQuery with regards to these. This paragraph gives a broader picture than the rest of the manuscript, and we hope that it does not feel off topic.

To avoid another "cingulategate", we also chose to be very explicit in the Discussion: "A NeuroQuery prediction by itself therefore does not support definite conclusions", and later: "What NeuroQuery does not do is provide conclusive evidence that a brain region is recruited by a mental process, or affected by a pathology".

B) For example, let's consider again the case about "combination of mental processes that have been studied separately, but never together". From my understanding of the algorithm, suppose users query "auditory" + "working memory", basically the prediction will be a linear combination of activation maps from "auditory" and "working memory" (+ similar terms due to the smoothing/query expansion). As such, this assumes that compound mental processes yield activations that are linear combination of activation maps of individual mental processes. This should be made clear.

Indeed this is our assumption, though it is a common one in functional neuroimaging (often referred to as the "pure insertion" hypothesis). We have made it very explicit in our own limitation paragraph and have exhibited a situation where it leads to undesirable result.

C) Playing around with NeuroQuery, there are some queries that generate obviously wrong results. For example, "autobiographical memory" should probably yield the default network, but we getting hippocampal/retrosplenial activation instead. This presumably happens because NeuroQuery "expanded" the query to become memory because "autobiographical memory" is not one of the 200 keywords? Interesting that NeuroSynth does get it correct (https://neurosynth.org/analyses/terms/autobiographical%20memory/).

Neuroquery’s map is actually quite close to neurosynth: both segment the hippocampus, the retrosplenial cortex and the precuneus. The difference lies mostly in the ACC/parietal parts of the default mode network. However we are aware that the query expansion is sometimes too sharp and can deteriorate results. In future work, we hope that ensemble models can help mitigate this issue. A first approach using random subspaces is already demonstrated in the gallery of neuroquery examples, and in an interactive demo, and seems to improve results for the case of "autobiographical memory".

The queries that give maps that do not ressemble what we would expect from an experiment targeted to answer the corresponding question are often queries that combine a term that is well mapped in brain-imaging experiments with a term that is harder to capture, such as "autobiographical memory of faces", as the published results are then dominated by "faces".

D) Perhaps a brief discussion of other limitations would be helpful. We submit that some of the fundamental problems are those not easily solved – that we usually perform meta-analyses based on studies of the same nominal task type (e.g., N-back), and sometimes minor variations in task structure can yield divergent findings. We don't know what all the dimensions are yet. This problem goes far beyond the challenge of establishing a set of consensus labels for task types and relevant cognitive processes. In short, we don't really even know what task features to label yet in many cases, and they don't combine additively. A stop-signal task with one adaptive random walk may be different than one with four, as it allows a different type of cognitive strategy.

We have included a discussion of the issue mentioned by the reviewers in the Discussion, starting with "Another fundamental challenge of meta-analyses in psychology is the decomposition of the tasks in mental processes […]"

E) When to use it: For common terms, meta-analysis (e.g., Neurosynth) does very well (e.g., Figure 4). When would the authors recommend using NeuroQuery over another meta-analytic tool? Maybe they could provide a summary of use cases and conditions (e.g., when few studies of a term/topic are available). Also see point (C).

We have added recommendations on when NeuroQuery is beneficial around, starting with "NeuroQuery will be most successfully used to produce hypotheses […]", and this paragraph finishes with a list of situations where using NeuroQuery is particularly indicated.

8) Discussion of other approaches: We submit that the field has become tracked into a relatively narrow space of the possible options and techniques for meta-analysis, based on local analysis of coordinates in MKDA/ALE. Alternatives could be mentioned as potential future directions. For example, early work explored clustering of spatial locations and spatial discriminant tests (e.g., Wager et al., 2002, 2004, 2005), and later work has explored spatial models (e.g., Kang, 2011, Kang, 2014, Wager, 2015) and more advanced co-activation models (Xue, 2014). While this is obviously beyond the scope of the present paper, future work might consider models of spatial co-activation when generating predictions and inferences about meta-analytic maps.

We now mention this line of work at the end of the paragraph on prior work, starting with "other works have modelled co-activations and interactions between brain locations"

One way to adapt the NeuroQuery model to better leverage the spatial structure of activations could be to use loadings on dictionary or ICA components as dependent variables rather than estimated density at each voxel.

9) The methods are a bit unclear:

We have made important changes to the organization and notations of the methods. Below we provide answers to the issues raised by the reviewers.

A) Significant details about methodologies are missing. How are term frequency and inverse document frequency computed?

We have added a section B.1 on the computation of TFIDF features.

*B) Equation 5: How is* σ^*_ij_ computed? Square root of the entries of equation 8?*

σ^2jk is now explicitly defined in Equation 9.

*C) Equation 5: What is the difference between* σ^*_i_ and* σ^*_ij_?* σ^*_i_ is a column/row of* σ^*_ij_?*

Indeed σ^*_j_* is the vector containing the entries (σ^_*j*_,1,..., σ^_*j*_,*p*) as made explicit at the end of section B.2.1. We have also added a short section on notations

D) Y:,j is the j-th column not i-th column of Y?

Indeed this was a typo, which has now been corrected (below Equation 6.)

*E) M is a v x n matrix, so equation 8 is a v x v matrix? We are confused how this maps to* σ^*_ij_.*

We have now defined σ^_*ij*_ in Equation 9 using the sum notation, we hope that this definition is now clear.

F) In Equation 10, is ||U||_1 just the sum of absolute values of all entries in U?

This is what we intended. This choice of notation was bad as it is usually employed for the operator norm induced by the l1 vector norm. We replaced this notation with that of the 𝐿_1,1_ norm ie ||.||_1,1_

G) What is k set to be in equation 10?

k has now been renamed d to avoid confusion with the index k used to index voxels. It is set to 300, as is now specified in the paragraph after Equation 18: "the hyperparameters d=300,.…"

H) "More than 72% of the time, NeuroQuery's output has a higher Pearson correlation with the correct map than with the negative example" – "correct map" refers to the KDE density maps?

"correct map" refers here to an unsmoothed maps where peaks have been placed in the voxels that contain the reported peak coordinates. We have now clarified this in the text "whether the predicted map is closer to the correct map (containing peaks at each reported location) or to the random negative example"

10) Subsection “Smoothing: regularization at test time” is hard to read. It would be helpful if the authors explain the intuition behind the different steps and what the different matrices represent. For example, it might be helpful to explain that each row of V can be thought of as the association of words with topic k, so a higher value for row i, column j suggests the j-th dictionary word is more strongly related to the topic k. As another example, the authors should also unpack subsection “Smoothing: regularization at test time”: why do we take V and scale with 𝑛_𝑖,𝑖_ and then compute C and then l1-normalizing the rows of C to produce T and then finally S. What does each step try to do? We guess roughly speaking VV^T is like a co-occurrence matrix (how likely are two words likely to appear together?), but we are not sure why we have to do the extra normalization with 𝑛_𝑖,𝑖_, l1-normalization, etc.

We have reworked this section in depth. More details on the interpretation of rows of 𝑉 are provided in the second paragraph after Equation 18, starting with "The latent factors, or topics, are the rows of.…"We provide some justification for scaling 𝑉 with 𝑛_𝑖,𝑖_ in equations 19-21: as the reviewers suspected, V~TV~~ 𝑉 𝑇 ~ 𝑉 can be seen as a regularized (low-rank) co-occurrence matrix. Our motivation for the subsequent l1-normalization is stated after Equation 22: "This normalization ensures that terms that have many neighbors are not given more importance in the smoothed representation". Overall we hope that this section is now clearer and provides better intuitions and justifications for the choices we made in the design of the NeuroQuery model.